# The evolution of non-reproductive workers in insect colonies with haplodiploid genetics

**Jason W Olejarz[1], Benjamin Allen[1,2,3], Carl Veller[1,4], Martin A Nowak[1,4,5]***

[1]Program for Evolutionary Dynamics, Harvard University, Cambridge, United States; [2]Center for Mathematical Sciences and Applications, Harvard University, Cambridge, United States; [3]Department of Mathematics, Emmanuel College, Boston, United States; [4]Department of Organismic and Evolutionary Biology, Harvard University, Cambridge, United States; [5]Department of Mathematics, Harvard University, Cambridge, United States

**Abstract** Eusociality is a distinct form of biological organization. A key characteristic of advanced eusociality is the presence of non-reproductive workers. Why evolution should produce organisms that sacrifice their own reproductive potential in order to aid others is an important question in evolutionary biology. Here, we provide a detailed analysis of the selective forces that determine the emergence and stability of non-reproductive workers. We study the effects, in situations where the queen of the colony has mated once or several times, of recessive and dominant sterility alleles acting in her offspring. Contrary to widespread belief based on heuristic arguments of genetic relatedness, non-reproductive workers can easily evolve in polyandrous species. The crucial quantity is the functional relationship between a colony's reproductive rate and the fraction of non-reproductive workers present in that colony. We derive precise conditions for natural selection to favor the evolution of non-reproductive workers.

**\*For correspondence:**
martin_nowak@harvard.edu

**Competing interests:** The authors declare that no competing interests exist.

## Introduction

Eusociality is a form of social organization where some individuals reduce their own lifetime reproductive potential to raise the offspring of others (*Wilson, 1971*; *Crespi and Yanega, 1995*; *Gadagkar, 2001*; *Hunt, 2007*; *Nowak et al., 2010*). Primary examples are ants, bees, social wasps, termites, and naked mole rats. There have been ~10–20 origins of eusociality, about half of them in haplodiploid species and the other half in diploid ones (*Andersson, 1984*). A crucial step in the origin of eusociality is cancellation of dispersal behavior (*Abouheif and Wray, 2002*; *Nowak et al., 2010*, *2011*; *Hunt, 2012*; *Tarnita et al., 2013*). Individuals who stay at the nest begin to work at tasks such as care of the young, defense of the nest, and foraging behavior. Eusociality can evolve if the strong reproductive advantages of the queen, including reduced mortality and increased rate of oviposition, arise already for small colony sizes (*Nowak et al., 2010*).

Haplodiploidy is the sex-determination system of ants, bees, and wasps. With haplodiploidy, females arise from fertilized eggs and are diploid. They have two homologous sets of chromosomes, one padumnal (paternally inherited) and one madumnal (maternally inherited). Males arise from unfertilized eggs and are haploid. They have one set of chromosomes, which is madumnal. With haplodiploid genetics, mated females can become queens and lay female and male eggs. In many cases, unmated females become workers, which can lay male eggs but not female eggs. Therefore, in a haplodiploid colony, male eggs could come from the queen or from the workers. This creates intracolony competition between the fertilized queen and the unfertilized workers over the production of

**eLife digest** Certain wasps, bees and ants live in highly organized social groups in which one member of a colony (the queen) produces all or almost all of the offspring. This form of social organization – called eusociality – raises an important question for evolutionary biology: why do individuals that forego the chance to reproduce and instead raise the offspring of others evolve?

One factor linked to the evolution of eusociality in insects is a system that determines the gender of offspring known as haplodiploidy. In this system, female offspring develop from fertilized eggs, while male offspring develop from unfertilized eggs. The queen mates with male insects and so she can produce both male and female offspring. On the other hand, the workers – which are also female – do not mate and therefore can only produce male offspring.

So, should these workers produce their own male eggs, or should all male offspring come from the queen? The answer to this question could depend on whether the queen has mated with a single male (monandry) or with multiple males (polyandry) because this affects how closely related the other insects in the colony are to each other. It is a widespread belief that monandry is important for the evolution of non-reproductive workers.

Here, Olejarz et al. develop a mathematical model that explores the conditions under which natural selection favors the evolution of non-reproductive workers. Contrary to the widespread belief, it turns out that non-reproductive workers can easily evolve in polyandrous species. The crucial quantity is the relationship between the overall reproductive rate of the colony and the fraction of non-reproductive workers present in that colony.

Olejarz et al. challenge the view that single mating is crucial for the evolution of non-reproductive workers. The study demonstrates the need for precise mathematical models of population dynamics and natural selection instead of informal arguments that are only based on considerations of genetic relatedness.

males. In this paper, we investigate genetic mutations that affect the phenotype of the workers and make them non-reproductive (or sterile). We calculate conditions both for the evolutionary invasion and evolutionary stability of such mutations.

The typical theoretical approach for investigating the evolution of non-reproductive workers makes use of coefficients of relatedness. Relatedness of one individual to another is the probability that a random allele in the former is also in the latter due to recent common ancestry. In particular, there are three coefficients of relatedness that are of primary interest (*Hamilton, 1964*; *Trivers and Hare, 1976*). First, consider the relatedness of a female to one of her sons, $R_{son}$. In parthenogenetically producing a son, a diploid female transmits a haploid genome to him. Under Mendelian segregation, each allele in her genome has probability 1/2 of inclusion in this transmitted haploid complement, and so $R_{son} = 1/2$. Next, consider the relatedness of a female to one of her brothers, $R_{brother}$. The female and her brother share a mother, and so the probability that an allele inherited maternally by the female is the same allele as inherited maternally by her brother is 1/2. On the other hand, the male has no father, and so there is no chance that an allele in the female is equivalent to one in her brother through paternal inheritance. A random allele in the diploid female has equal chance of being padumnal or madumnal, and so $R_{brother} = (1/2)(0) + (1/2)(1/2) = 1/4$. Finally, consider the relatedness of a female to one of her sisters, $R_{sister}$. This coefficient of relatedness depends on the number, $n$, of different males that the queen mates with before laying eggs. For a padumnal allele in a female to be identical, by paternal descent, to that in a sister requires them only to share the same father (probability $1/n$), since that father is haploid and therefore always transmits the same allele. Sisters share a mother, and so the probability that an allele inherited maternally by one female is the same allele as inherited maternally by her sister is 1/2. Therefore, $R_{sister} = (1/2)(1/n) + (1/2)(1/2) = (2 + n)/(4n)$.

The traditional investigation of evolution of non-reproductive workers uses the following relatedness-based heuristic. The relatedness of a female to her male offspring is $R_{son} = 1/2$. If the queen mates with only a single male ($n = 1$), then the relatedness of a female worker to one of her random sisters is $R_{sister} = 3/4 > R_{son}$. The naive conclusion is that worker altruism should readily evolve,

because a worker can more efficiently spread her genes by raising her sisters. This conjecture is known as the 'haplodiploidy hypothesis' (*Hamilton, 1964*, *1972*). If the queen mates with more than two males ($n>2$), then $R_{sister} < R_{son}$. Now the preference of an unfertilized worker is reversed, because she has a higher relatedness to her own male offspring than to one of her random sisters. These old arguments suggest that queen monogamy and haplodiploid genetics synergistically act as a driving force for the evolution of worker altruism (*Hamilton, 1964*, *1972*).

But worker-laid eggs do not compete only with queen-laid female eggs. The high relatedness of a female to her sisters in Hymenopteran colonies is cancelled by the low relatedness of the same female to her brothers (*Trivers and Hare, 1976*). In a colony with a singly mated queen, the relatedness of a worker to a sister is $R_{sister} = 3/4$. But the relatedness of a worker to a brother is only $R_{brother} = 1/4$, regardless of the number of times the queen mates. So, when a worker female helps her queen reproduce, she aids in the production both of sisters (to whom she is highly related) and brothers (to whom she is not). In the relatedness-based argument, these effects exactly cancel each other out, and so the unusually high relatedness of sisters in eusocial colonies cannot be the simple solution to the puzzle of worker altruism that it was once thought to be (*Trivers and Hare, 1976*). This is true even when the population sex ratio is female-biased, because when more reproductive females are produced than males, the average reproductive success of a female is lower than that of the average male exactly in proportion to their relative abundances (*Craig, 1979*).

More recently, it was proposed that each eusocial lineage must have passed through a 'monogamy window'—a period of evolutionary history in which queens were singly mated (*Boomsma, 2009*). This argument assumes that worker-laid eggs compete equally with queen-laid female and male eggs. If the queen is singly mated ($n = 1$), and if the colony's sex ratio is 1/2, then a worker has an average relatedness to siblings (sisters and brothers) of ($R_{sister} + R_{brother}$)/2 = ((3/4) + (1/4))/2 = 1/2. Relatedness of a worker to her son is also $R_{son} = 1/2$. In this case, assuming that worker-laid eggs substitute equally for queen-laid female and male eggs, the argument suggests that any infinitesimal benefit of non-reproductive workers to colony productivity should lead to evolution of worker altruism. If queens mate more than once ($n \geq 2$), then the average relatedness of a worker to a random sibling falls below 1/2, and evolution of worker altruism is supposed to be strongly disfavored (*Boomsma, 2009*).

There also exist hypothetical scenarios in which the relatedness values of a female to a random sister and a random brother would not cancel (*Trivers and Hare, 1976*): for example, the sex ratio could vary from colony to colony, while the average sex ratio in the population remains at 1/2. Evolution of helping might then be expected in the female-biased colonies (*Trivers and Hare, 1976*). Some recent papers (*Gardner et al., 2012*; *Alpedrinha et al., 2013*) examine this case of split sex ratios and also question the importance of haplodiploidy for the evolution of helping. Based on their analysis, the authors conclude that haplodiploidy can have either a positive or negative influence on the evolution of helping depending on colony variables, and they determine the effect of haplodiploidy on the evolution of helping to generally be small. But they claim, in agreement with *Boomsma (2009)*, that monandry is a key requirement for the evolution of a worker caste. They argue that, in the case of lifetime monogamy, "any small efficiency benefit from rearing siblings ($b/c > 1$) would lead to helping being favored by natural selection." (*Gardner et al., 2012*)

In this paper, we investigate the situation where worker-laid male eggs compete directly with queen-laid male eggs. In other words, there are two types of reproduction events to consider: A worker can lay a male egg, or, instead, the queen can lay a male egg while the worker helps to raise the queen's male egg. In either case, a male is produced. Thus, the sex ratio of the colony is unaffected by which of these two strategies is realized. The reproductive competition between the queen and the workers is only over the male offspring. (We do not assume that the sex ratio is equal to 1/2; we only assume that it is independent of the fraction of worker-produced males.) This possibility represents the simplest scenario for studying the evolution of non-reproductive workers and is biologically plausible (*Winston, 1987*; *Sundstrom, 1994*; *Sundstrom et al., 1996*; *Queller and Strassmann, 1998*; *Hammond et al., 2002*). Once this case is understood, subsequent analysis can consider the situation where both the fraction of worker-produced male offspring and the colony's sex ratio vary at the same time.

The relatedness of a worker to her male offspring, $R_{son} = 1/2$, is always larger than the relatedness of a worker to a brother, $R_{brother} = 1/4$. Thus, if worker-laid male eggs compete primarily with queen-laid male eggs, then relatedness-based arguments might predict that worker altruism should

not evolve with any number of matings of the queen unless non-reproductive workers provide some benefit to the colony.

We study a model of competition between worker-laid and queen-laid male eggs that incorporates both haplodiploid genetics and a variable number of matings of the queen. Specifically, we analyze the selection pressure acting on the emergence and stability of non-reproductive workers in situations where the queen has mated once or several times. Our model assumes that the evolution of sterile workers has a negligible effect on the sex ratio of a population, which allows us to isolate and identify the specific selective forces. We derive exact conditions for the invasion and stability of non-reproductive workers.

## Model

We study the evolution of a non-reproductive worker caste in the context of haplodiploid genetics, where females are diploid and males are haploid. Virgin queens can mate with one or several males. The parameter $n$ denotes the number of matings of the queen. Unfertilized workers help raise the offspring of the queen, but they can also lay male eggs.

We analyze the conditions under which a wild-type allele, $A$, can be invaded by a mutant allele, $a$, which causes workers to be non-reproductive. Since we consider a loss of function event (the loss of the tendency to produce eggs), it is more likely that the mutation is recessive rather than dominant. Therefore in the main text we present the conditions for a recessive mutant allele. In the Methods, we give derivations and results for both recessive and dominant alleles.

If the mutant allele is recessive, then $aa$ workers are sterile, while $AA$ and $Aa$ workers still lay male eggs. For $n$ matings, there are $3(n + 1)$ types of mated queens (*Figure 1A*). We use the notation $AAm$, $Aam$, and $aam$ to denote the genotype of the queen and the number, $m$, of her matings that were with mutant males, $a$. The parameter $m$ can assume values 0, 1, ..., $n$. For example, for triple mating ($n = 3$), an $AA2$ queen has mated with one $A$ male and two $a$ males.

The genotype of the colony is determined by the genotype of the queen and the sperm she has stored. There are $3(n + 1)$ types of colonies that need to be considered to formulate the full dynamics. The different colony types, and corresponding offspring with a recessive sterility allele, $a$, are shown in *Figure 1B*.

For each colony, there are three types of offspring: queen-laid females, queen-laid males, and worker-laid males. *Figure 1B* can be understood by considering how the queen and the workers produce their offspring.

Consider the offspring of type $AAm$ colonies. The queen makes a female by randomly selecting one of the two alleles from her own genotype and pairing it with an allele selected randomly from the sperm of one of her mates. In type $AAm$ colonies, the type $AA$ queen mates with ($n - m$) type $A$ males and $m$ type $a$ males. From her own genotype, the queen always selects an $A$ allele. From her mates' sperm, the queen selects an $A$ allele with probability ($n - m$)/$n$ or an $a$ allele with probability $m/n$. Notice that in *Figure 1B,C*, for simplicity of presentation, we omit the overall normalization of each entry. So, for example, for type $AAm$ colonies, we simply write that the queen produces $n - m$ type $AA$ females for every $m$ type $Aa$ females (first row, second column of *Figure 1B*). The correct normalizations are included in the calculations of the Materials and methods section.

The queen makes a male by randomly selecting one of the two alleles from her own genotype. Because the queen in a type $AAm$ colony only carries the $A$ allele, she can only produce type $A$ drones (first row, third column of *Figure 1B*).

For a type $AAm$ colony, in the 'Workers' Sons' column, we must consider the rates of production of type $AA$ and type $Aa$ females by the queen, and we must consider the offspring of the type $AA$ and type $Aa$ females that the queen produces. The fraction of queen-produced females of type $AA$ is, as described above, ($n - m$)/$n$, and each type $AA$ female produces only type $A$ males. The fraction of queen-produced females of type $Aa$ is, as described above, $m/n$, and each type $Aa$ female produces type $A$ and type $a$ males with equal probability. The total fraction of worker-laid males that are of type $A$ is ($n - m$)/$n$ + (1/2)($m/n$) = ($2n - m$)/$2n$. The total fraction of worker-laid males that are of type $a$ is (1/2)($m/n$) = $m/2n$. Therefore, the workers of type $AAm$ colonies produce $2n - m$ type $A$ males for every $m$ type $a$ males (first row, fourth column of *Figure 1B*).

The logic behind the offspring of type $Aam$ and type $aam$ colonies is the same. The only other point is that type $aa$ workers are non-reproductive. To see how worker sterility enters into

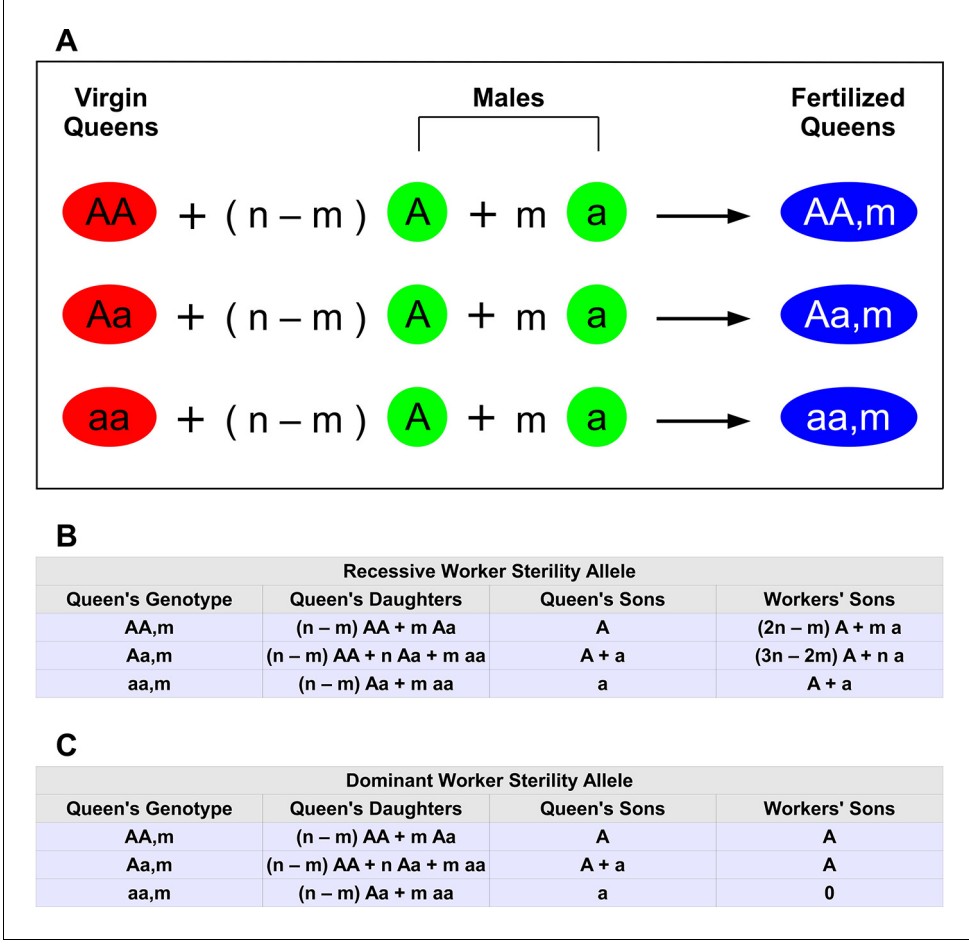

**Figure 1.** Haplodiploid genetics and multiple matings. The wild-type allele is *A*. The mutant allele inducing worker sterility is *a*. (**A**) There are three types of virgin queens: *AA*, *Aa*, and *aa*. Each queen mates *n* times. Of those matings, $n - m$ are with wild-type males (type *A*) and *m* are with mutant males (type *a*). Hence, there are $3(n + 1)$ types of fertilized queens (colonies). (**B**) Relative proportions of offspring for each colony type if the mutant allele, *a*, for worker sterility is recessive. For example, if the queen's genotype is *Aa*, then half of her sons are *A* and the other half are *a*. We denote this by $A + a$. If the queen's genotype is *aa* and she has mated with both types of males, $0 < m < n$, then she has both *Aa* and *aa* workers (in proportion $n - m$ and *m*, respectively); her *Aa* workers produce male eggs, which have an equal proportion of *A* and *a* genotypes. (**C**) Relative proportions of offspring for each colony type if the mutant allele, *a*, for worker sterility is dominant.

*Figure 1B*, consider the worker-produced males of type *aam* colonies. The queen of type *aam* colonies produces $n - m$ type *Aa* females for every *m* type *aa* females (third row, second column of *Figure 1B*). Type *Aa* workers produce equal numbers of type *A* and type *a* males. Type *aa* workers are non-reproductive; they do not contribute to the colony's production of worker-produced males. So, in type *aam* colonies, all worker-produced males come from type *Aa* workers, and type *A* and type *a* males are therefore produced by workers in equal amounts (third row, fourth column of *Figure 1B*).

The entries in *Figure 1C* originate from the same reasoning. The only difference is that, if the sterility allele, *a*, is dominant, then type *Aa* and type *aa* workers do not contribute to a colony's production of worker-produced males. The entries in *Figure 1C* are described in detail in the Materials and methods.

In our analysis, we neglect stochastic effects. This is reasonable if we assume that the number of individuals produced by a colony is very large. In this case, the fractions of colony offspring of different genotypes in a generation do not differ significantly from the entries in *Figure 1B,C*.

A crucial quantity is the functional relationship between the fraction of males produced by the queen, $p$, and the fraction of non-reproductive workers, $z$, that are present in a colony. The parameter $z$ can vary between 0 and 1. If $z = 0$, then there are no non-reproductive workers in the colony. If $z = 1$, then all workers in the colony are non-reproductive. We denote by $p_z$ the fraction of males that come from the queen if the fraction of non-reproductive workers is $z$. The quantity $p_0$ denotes the fraction of males that come from the queen if there are no non-reproductive workers in the colony. We expect $p_0$ to be less than 1. The quantity $p_1$ denotes the fraction of males that come from the queen if all workers are non-reproductive. Clearly, $p_1 = 1$.

It is natural to assume that $p_z$ is an increasing function of $z$, but various functional forms are possible. Perhaps the simplest possibility is that $p_z$ is a linearly increasing function of $z$. Intuitively, this means that the fraction, $1 - p_z$, of male eggs that originate from workers is simply proportional to the fraction, $1 - z$, of workers that are reproducing. But there are nonlinear intracolony effects that modulate worker production of male eggs. For example, the queen might efficiently suppress worker reproduction via aggression or removal of worker-laid eggs (*Free and Butler, 1959*; *Michener, 1974*; *Oster and Wilson, 1978*; *Fletcher and Ross, 1985*), if only a small number of workers attempt to reproduce. If too many workers reproduce, then the queen could be overwhelmed, and her effect on removing worker-laid eggs is diminished. In this equally plausible scenario, the fraction, $p_z$, of male eggs that originate from the queen would be expected to increase sublinearly with the fraction, $z$, of workers that are sterile. Several sample forms of the function $p_z$ are shown in *Figure 2A*.

The mutant allele can be favored by natural selection if non-reproductive workers provide a benefit to the colony, which is of course a natural assumption for the evolution of worker altruism. Division of labor has the potential to improve efficiency (*Cole, 1986*; *Naeger et al., 2013*). Another key component in our analysis is the functional relationship between the rate, $r$, at which the colony produces reproductive units (virgin queens and males) and the fraction of sterile workers, $z$. We use the notation $r_z$ to describe the reproductive rate of a colony where a fraction, $z$, of workers are non-reproductive. The quantity $r_0$ denotes the reproductive rate of the colony if none of the workers are non-reproductive. Non-reproductive workers have a chance to be favored by natural selection if

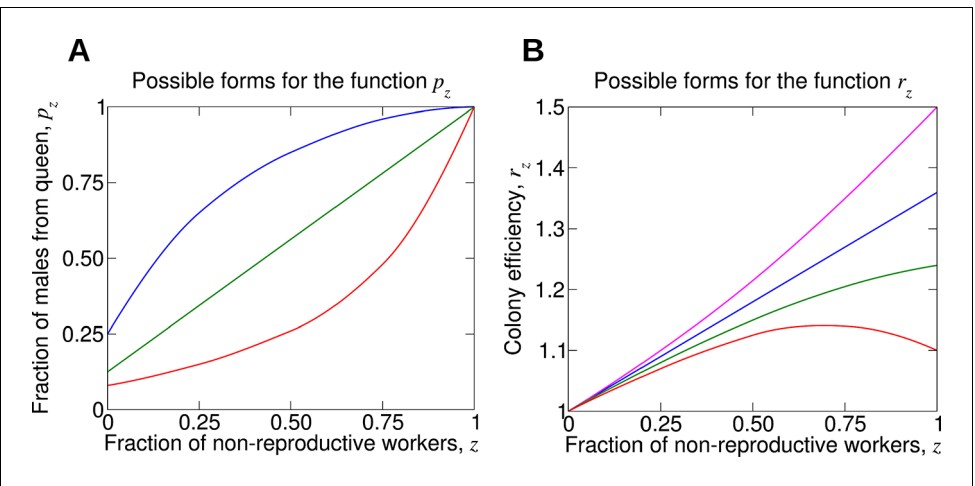

**Figure 2.** For understanding the evolution of non-reproductive workers, the following two functions are crucial. (A) The function $p_z$ denotes the fraction of male offspring that come from the queen if a fraction, $z$, of the workers are non-reproductive. Therefore, $1 - p_z$ is the fraction of male offspring that come from the workers. Clearly, $p_z$ should be an increasing function. More workers that are sterile means a larger fraction of males that come from the queen. If all workers are non-reproductive, then all males come from the queen, $p_1 = 1$. (B) The function $r_z$ denotes the reproductive rate (or efficiency) of the colony if a fraction, $z$, of the workers are non-reproductive. Without loss of generality, we normalize such that $r_0 = 1$. If worker sterility has an advantage, then it should increase colony efficiency for some values of $z$, but the function $r_z$ need not be monotonically increasing. It is possible that maximum colony efficiency is obtained for an intermediate value of $z$. Several possibilities for the colony efficiency function, $r_z$, are shown.

$r_z > r_0$ for some $z$. But the function $r_z$ need not be monotonically increasing. It is possible that there is an optimum fraction of non-reproductive workers, which maximizes the overall reproductive rate of the colony. We will study various functional forms of $r_z$. Several sample forms of the function $r_z$ are shown in *Figure 2B*.

## Results

If the mutant allele is recessive, then *AA* and *Aa* workers lay male eggs, while *aa* workers are non-reproductive. For single mating, $n = 1$, we find that the *a* allele can invade an all-*A* resident population provided

$$\frac{r_{1/2}}{r_0} > \frac{6 + 2p_0}{5 + 3p_0} \tag{1}$$

What is the intuition behind this condition? There are five colony types, *AA*0, *AA*1, *Aa*0, *Aa*1, and *aa*0, which are relevant for determining if the mutant allele can invade. Four of those colony types do not produce sterile workers ($z = 0$), so the parameters $p_0$ and $r_0$ enter into *Equation 1*. In colonies of type *Aa*1, half of the workers are sterile ($z = 1/2$); thus the parameter $r_{1/2}$ enters into *Equation 1*. Moreover, both the queen and the workers in *Aa*1 colonies each produce 50% type *A* males and 50% type *a* males; therefore the parameter $p_{1/2}$ is irrelevant for the invasion and absent from *Equation 1*.

If all males are initially produced by the workers ($p_0 = 0$), then the ratio of the efficiency of type *Aa*1 colonies to type *AA*0 colonies, $r_{1/2}/r_0$, must be greater than 6/5 for non-reproductive workers to appear. Notice that the critical value of $r_{1/2}/r_0$ is a decreasing function of $p_0$. Intuitively, this means that if worker sterility has a smaller phenotypic effect on a colony (such that $p_0$ is closer to $p_1 = 1$), then the efficiency gain from sterile workers does not need to be as high to facilitate the invasion of sterility. If $p_0$ is small, then we get efficiency thresholds for $r_{1/2}/r_0$ of ~1.1-1.2. If $p_0$ is large, then we get efficiency thresholds that are close to 1. As long as $p_0$ is not infinitesimally close to 1, the ratio $r_{1/2}/r_0$ must always be greater than 1 by a finite amount. Sterility cannot invade if sterile workers do not appreciably improve colony efficiency.

We note that other studies report the evolution of a worker caste with infinitesimal efficiency benefits in singly mated colonies (*Boomsma, 2007*, *2009*; *Gardner et al., 2012*). But these papers consider competition between worker offspring and queen-laid female eggs, which induces sex-ratio effects that complicate the analysis.

Another recent study argues that eusociality can evolve even if sterile workers are relatively inefficient at raising siblings (*Avila and Fromhage, 2015*). But this work focuses centrally on the evolution of nest formation, where nest-site limitation and dispersal mortality impose ecological constraints on independent breeding. In our study, we analyze the scenario where nests have already formed, and non-reproductive workers emerge as a subsequent step in the path to advanced eusociality.

For double mating, $n = 2$, we find that the *a* allele can invade an all-*A* resident population provided

$$\frac{r_{1/4}}{r_0} > \frac{6 + 3p_0}{5 + 3p_0 + p_{1/4}} \tag{2}$$

Now there are six colony types, *AA*0, *AA*1, *AA*2, *Aa*0, *Aa*1, and *aa*0, which are relevant for determining if the mutant allele can invade. Colony types *AA*0, *AA*1, *AA*2, *Aa*0, and *aa*0 do not produce sterile workers ($z = 0$), so the parameters $p_0$ and $r_0$ appear in *Equation 2*. Colonies of type *Aa*1 produce a fraction 1/4 of sterile workers ($z = 1/4$). The *Aa*1 queen uses the sperm from the type *A* male that she has mated with to produce *AA* and *Aa* workers in equal proportion. Or the *Aa*1 queen uses the sperm from the type *a* male to produce *Aa* and *aa* workers with equal proportion. Thus, 1/4 of the workers are of type *aa* and are non-reproductive. Correspondingly, the parameters $p_{1/4}$ and $r_{1/4}$ appear in *Equation 2*.

The maximum critical value of $r_{1/4}/r_0$ for evolution of non-reproductive workers is 6/5, and the minimum critical value is 1. The threshold of $r_{1/4}/r_0$ is large (~1.1-1.2) if $p_0$ is small. The threshold of $r_{1/4}/r_0$ is close to 1 if $p_0$ is large. Provided that $p_0$ is not infinitesimally close to 1, the ratio $r_{1/4}/r_0$ must always be greater than 1 by a finite amount for sterile workers to be able to invade.

It is not clear, a priori, that *Equation 2* would be easier or harder to satisfy than *Equation 1*. Empirical knowledge of the parameters $p_0$, $p_{1/4}$, $r_0$, $r_{1/4}$, and $r_{1/2}$ is needed to determine whether sterility invades more easily for single mating than for double mating.

An illustration of the parameter space and whether single or double mating is more conducive to development of sterility is shown in *Figure 3A*. It is clear from *Equation 1* that, holding all other parameters constant, an increase in $r_{1/2}$ favors the invasion of the sterility allele for $n = 1$. This is easy to see in *Figure 3A*: The upper panels (higher $r_{1/2}$) involve invasion of the sterility allele for $n = 1$, while the lower panels do not. Similarly, from *Equation 2*, it is clear that, holding all other parameters constant, an increase in $r_{1/4}$ favors the invasion of the sterility allele for $n = 2$. Again, this is illustrated in *Figure 3A*: the right panels (higher $r_{1/4}$) are associated with invasion of the sterility allele for $n = 2$, while the left panels are not.

The region of parameter space for which sterility invades for double mating but not for single mating is arbitrarily large. The region of parameter space for which sterility invades both for double mating and for single mating is also arbitrarily large. These features apply generally for different values of $p_0$ and $p_{1/4}$.

For many possible combinations of those parameters, worker sterility invades for double mating but not for single mating. For example, if $p_0 = 0.8$ and $p_{1/4} = 0.9$, then for single mating the invasion condition is $r_{1/2} > 1.027$ while for double mating the invasion condition is $r_{1/4} > 1.012$. (Here, without loss of generality, we set $r_0 = 1$.) The latter condition could be easier to satisfy—even if $r_z$ increases linearly with $z$.

We note that colony reproductive efficiency, $r_z$, would not necessarily be expected to increase monotonically with the fraction of sterile workers, $z$. The law of diminishing returns may apply to the addition of non-reproductive workers to a colony. Non-reproductive workers contribute positively to the colony's total reproductive output by performing colony maintenance and helping to raise other individuals' offspring. But by not laying any eggs, non-reproductive workers are also negatively affecting the colony's total reproductive output. Consequently, colony reproductive efficiency may be maximized if some workers reproduce while other workers focus their efforts on colony maintenance. In our model, this would correspond to $r_z$ reaching a maximum for some $0 < z < 1$.

Assuming that $p_0$ and $p_{1/4}$ are small, we find that a fairly substantial benefit to colony reproductive rate (around 10% to 20%) must be provided by a non-reproductive worker caste. The large thresholds predicted by our model might help to explain the rarity of the evolution of non-reproductive worker castes in social insects. Additional work is needed to connect the parameters of our

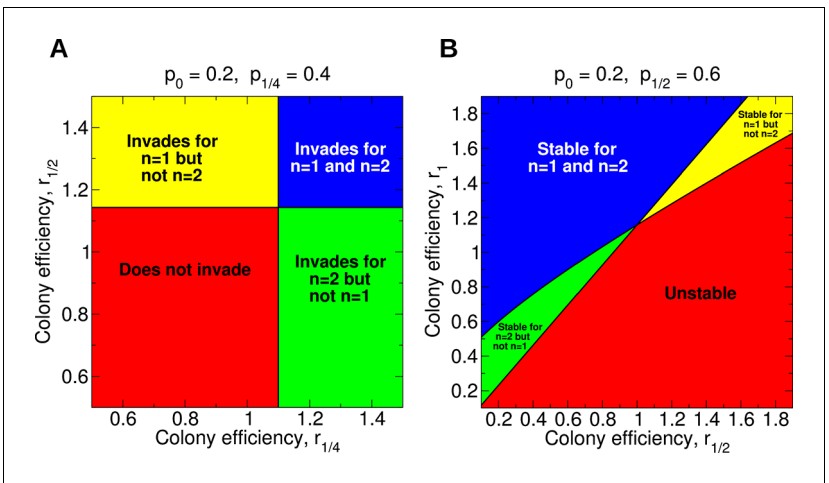

**Figure 3.** Regions of the parameter space for the evolution of non-reproductive workers for single and for double mating. (A) For single mating, $n = 1$, the invasion of a recessive worker sterility allele depends on the parameters $p_0$ and $r_{1/2}$; for double mating, $n = 2$, it depends on the parameters $p_0$, $p_{1/4}$, and $r_{1/4}$. (B) The evolutionary stability of a recessive worker sterility allele depends on the parameters $p_0$, $r_{1/2}$, and $r_1$ for single mating, and on the parameters $p_{1/2}$, $r_{1/2}$, and $r_1$ for double mating. We set $r_0 = 1$ as baseline.

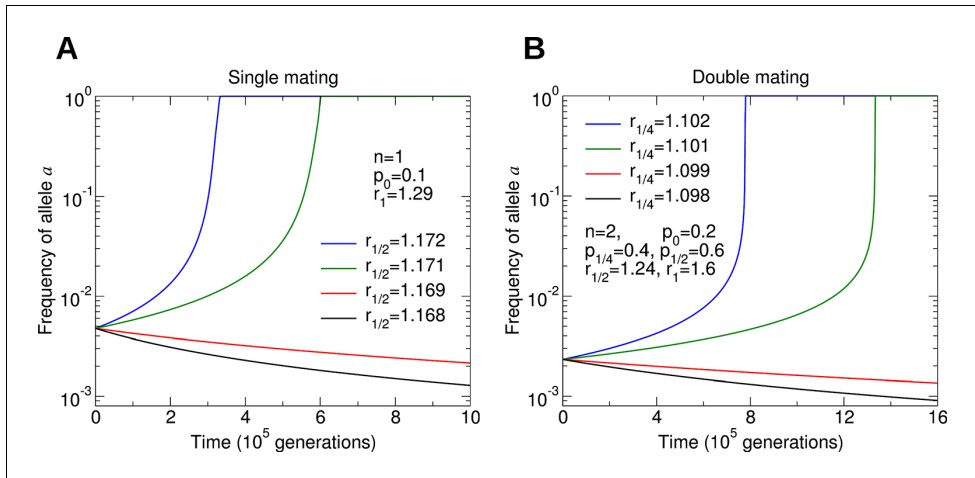

**Figure 4.** Numerical simulations of the evolutionary dynamics nicely illustrate the conditions specified by *Equations 1 and 2*. The sterility allele is recessive. For numerically probing invasion, we use the initial condition $X_{AA,0} = 1 - 10^{-2}$ and $X_{AA,1} = 10^{-2}$. We set $r_0 = 1$. A: Single mating, $n = 1$. Parameters $p_0 = 0.1$ and $r_1 = 1.29$. B: Double mating, $n = 2$. Parameters $p_0 = 0.2$, $p_{1/4} = 0.4$, $p_{1/2} = 0.6$, $r_{1/2} = 1.24$ and $r_1 = 1.6$.

model with biological measurements of colony dynamics. Numerical simulations of the evolutionary dynamics for different parameter values are shown in *Figure 4*.

We have also calculated the condition for the evolutionary stability of non-reproductive workers. For single mating, $n = 1$, we find that the *a* allele is stable against invasion of *A* in an all-*a* resident population provided

$$\left[ \frac{r_1}{r_0} - \frac{1 - p_0}{2} \right] \left[ 2 \left( \frac{r_1}{r_{1/2}} \right) - 1 \right] > 1 \tag{3}$$

Three colony types, *aa*1, *aa*0, and *Aa*1, are relevant for determining if the *a* allele for sterility is evolutionarily stable to invasion by the *A* allele. Type *aa*1 colonies produce only sterile workers ($z = 1$), hence the appearance of $r_1$ in *Equation 3*. Type *aa*0 colonies produce no sterile workers ($z = 0$), hence the appearance of $p_0$ and $r_0$ in *Equation 3*. Type *Aa*1 colonies produce 50% sterile workers ($z = 1/2$), hence the appearance of $r_{1/2}$ in *Equation 3*. The parameter $p_{1/2}$ is irrelevant because the queen and the reproductive workers in type *Aa*1 colonies each produce 50% type *A* males and 50% type *a* males.

For double mating, $n = 2$, we find that the *a* allele is evolutionarily stable provided

$$\left[ \frac{r_1}{r_{1/2}} - (1 - p_{1/2}) \right] \left[ 2 \left( \frac{r_1}{r_{1/2}} \right) - 1 \right] > 1 \tag{4}$$

Three colony types, *aa*2, *aa*1, and *Aa*2, are relevant for determining if the *a* allele for sterility is evolutionarily stable. Type *aa*2 colonies produce only sterile workers ($z = 1$), hence the appearance of $r_1$ in *Equation 4*. Type *aa*1 and type *Aa*2 colonies each produce 50% sterile workers ($z = 1/2$), hence the appearance of $p_{1/2}$ and $r_{1/2}$ in *Equation 4*. The conditions for invasion and stability with more than two matings are given in the Materials and methods.

Empirical knowledge of the parameters $p_0$, $p_{1/2}$, $r_0$, $r_{1/2}$, and $r_1$ is needed to determine if worker sterility is more stable for single mating than for double mating. For many possible combinations of those parameters, worker sterility is evolutionarily stable for double mating but not for single mating. For example, if $p_0 = 0.6$, $p_{1/2} = 0.9$, $r_0 = 1$, and $r_{1/2} = 1.05$, then for single mating the stability condition is $r_1 > 1.105$ while for double mating the stability condition is only $r_1 > 1.087$. The latter condition is less stringent. The parameter space for evolutionary stability for specific values of $p_z$ is shown in *Figure 3B*.

*Equations 1–4* tell us how non-reproductive workers evolve in a population of otherwise reproductive workers. The simplest case of singly mated queens already shows rich behavior. In *Figure 5A*, the four possibilities are shown: Sterility may not invade and be unstable (lower left),

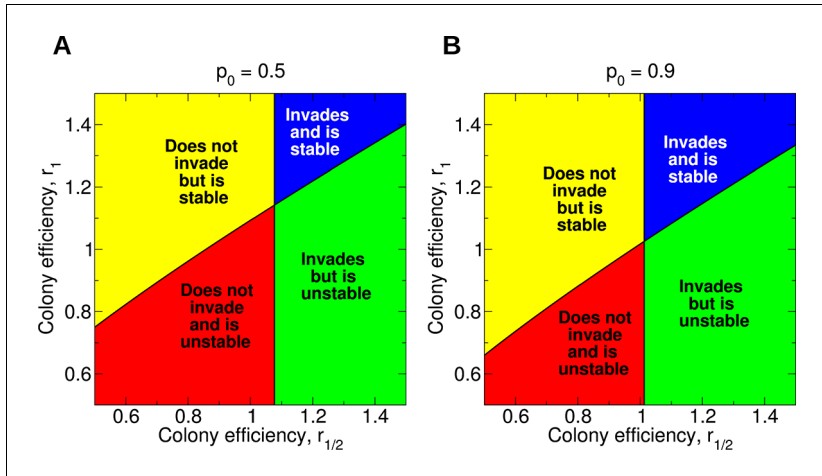

**Figure 5.** Evolution of non-reproductive workers for single mating ($n = 1$). We consider a recessive sterility allele, $a$. There are four possible scenarios: The mutant allele cannot invade but is evolutionarily stable (bistability); the mutant allele can invade and is evolutionarily stable; the mutant allele can invade but is unstable (coexistence); the mutant allele cannot invade and is unstable. Only three parameters matter: $p_0$, $r_{1/2}$, and $r_1$; $p_0$ denotes the fraction of male offspring that come from the queen if there are no sterile workers in the colony ($z = 0$); $r_{1/2}$ and $r_1$ denote respectively the reproductive rate (efficiency) of the colony if $z = 1/2$ and $z = 1$ of all workers are sterile. The baseline value is $r_0 = 1$. (**A**) Phase diagram for $p_0 = 0.5$. (**B**) Phase diagram for $p_0 = 0.9$. As $p_0$ gets closer to 1, the intersection of the critical curves approaches the point $(r_{1/2}, r_1) = (1,1)$.

invade but be unstable (lower right), not invade but be stable (upper left), or invade and be stable (upper right). For example, notice that if $r_{1/2} = 0.6$ and $r_1 = 0.9$, then worker sterility does not invade but is evolutionarily stable, even though both efficiency parameters are less than 1. As another example, notice that as long as $r_{1/2}$ exceeds about 1.077, the quantity $r_1$ can be arbitrarily small and worker sterility will still invade. It is also interesting that, for a fixed value of $r_1$, increasing the value of $r_{1/2}$ does not necessarily promote the stability of worker sterility, and doing so can actually render non-reproductive workers evolutionarily unstable. Complexities such as these are not readily accounted for by heuristic relatedness-based arguments. If the value of $p_0$ is very close to 1, then arbitrarily small changes in colony efficiency can positively or negatively influence the evolutionary invasion or stability of worker sterility (*Figure 5B*). Numerical simulations of the evolutionary dynamics demonstrating the four possible behaviors are shown in *Figure 6*.

*Figure 7* shows some examples. In *Figure 7A*, worker sterility invades for double mating but not for single mating. Here, $r_z$ increases sublinearly in $z$. In *Figure 7B*, the value of $p_{1/4}$ is only slightly increased compared with its value in *Figure 7A*. In *Figure 7B*, the efficiency function, $r_z$, is linearly increasing, and worker sterility invades for double mating but not for single mating. For the parameter values in *Figure 7C*, worker sterility is stable for double mating but not for single mating. Here, $r_z$ increases somewhat faster than linearly in $z$. In *Figure 7D*, the value of $p_{1/2}$ is only slightly increased compared with its value in *Figure 7C*. In *Figure 7D*, the efficiency function, $r_z$, is linearly increasing, and worker sterility is stable for double mating but not for single mating.

For a dominant sterility allele, there is typically a large region of parameter space for which sterility evolves for double mating but not for single mating. There is also an arbitrarily large region of parameter space for which sterility evolves for double mating and for single mating. For a recessive allele, it is possible that more than two matings are necessary for the emergence of worker sterility. These additional examples are presented in the Materials and methods.

## Discussion

Single mating of queens has often been claimed to be a key factor in the evolution of eusociality. This paradigm derives from heuristic relatedness-based arguments. For example, Boomsma's 'monogamy window hypothesis' (*Boomsma, 2007*, *2009*) holds that persistent monandry is crucial

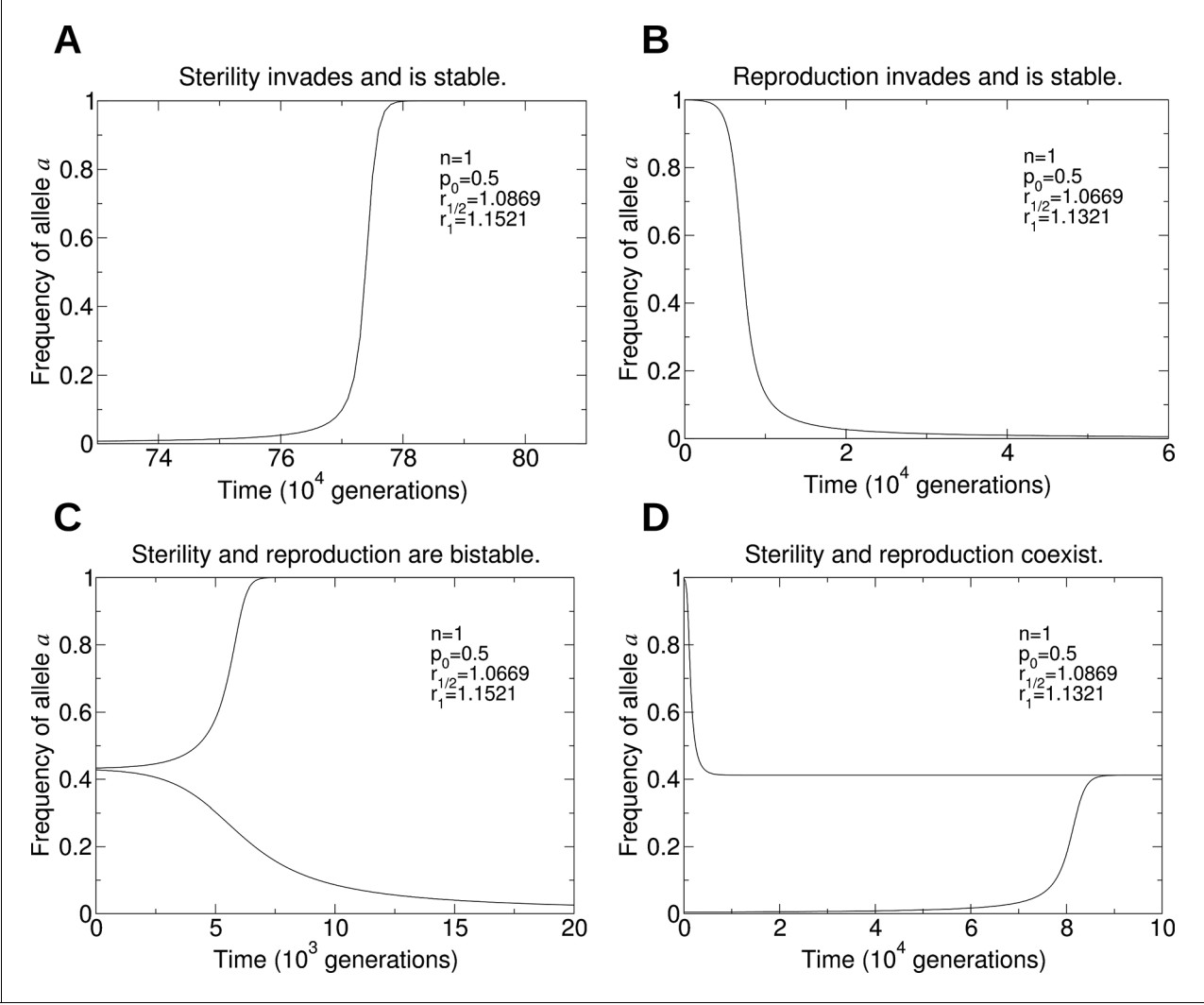

**Figure 6.** Numerical simulations of the evolutionary dynamics that show the four behaviors in *Figure 5A*. The sterility allele is recessive, and we consider single mating ($n = 1$). For each of the four panels, we use the initial conditions: A: $X_{AA,0} = 1 - 10^{-3}$ and $X_{AA,1} = 10^{-3}$; B: $X_{aa,1} = 1 - 10^{-3}$ and $X_{aa,0} = 10^{-3}$; C: $X_{AA,0} = 0.27$ and $X_{aa,0} = 0.73$ (lower curve), and $X_{AA,0} = 0.26$ and $X_{aa,0} = 0.74$ (upper curve); D: $X_{AA,0} = 1 - 10^{-2}$ and $X_{AA,1} = 10^{-2}$ (lower curve), and $X_{aa,1} = 1 - 10^{-2}$ and $X_{aa,0} = 10^{-2}$ (upper curve). We set $r_0 = 1$. A: Parameters $p_0 = 0.5$, $r_{1/2} = 1.0869$, and $r_1 = 1.1521$. B: Parameters $p_0 = 0.5$, $r_{1/2} = 1.0669$, and $r_1 = 1.1321$. C: Parameters $p_0 = 0.5$, $r_{1/2} = 1.0669$, and $r_1 = 1.1521$. D: Parameters $p_0 = 0.5$, $r_{1/2} = 1.0869$, and $r_1 = 1.1321$.

in the evolution of a worker caste. His argument is that, under monandry, because a worker is equally related to her own offspring as to her mother's offspring (1/2), her reduced reproduction will be selected for if it increases production of the latter more than it decreases production of the former. By this argument, under monandry but not multiple mating, very small colony-level efficiency gains from workers should lead to their evolution.

On the empirical side, *Hughes et al. (2008)* study a phylogeny of the Hymenoptera, and, employing an ancestral state reconstruction analysis, infer that each of the eight independent transitions to eusociality in the Hymenoptera occurred in a monandrous ancestral species. They take this correlation to be evidence for the causal claim that monandry is key to the evolution of eusociality. But, if most ancestral species in the clade were monandrous [as appears to be the case: *Hughes et al. (2008)*, Fig S1; *Nonacs (2011)*], then the fact that the ancestors of eusocial species in the clade were monandrous would not be surprising (*Nonacs, 2011*). As an extreme example, if Hughes et al. were to repeat their study for the trait of haplodiploidy, they would also find that each of the eight independent transitions to eusociality occurred in a haplodiploid ancestral species (since all

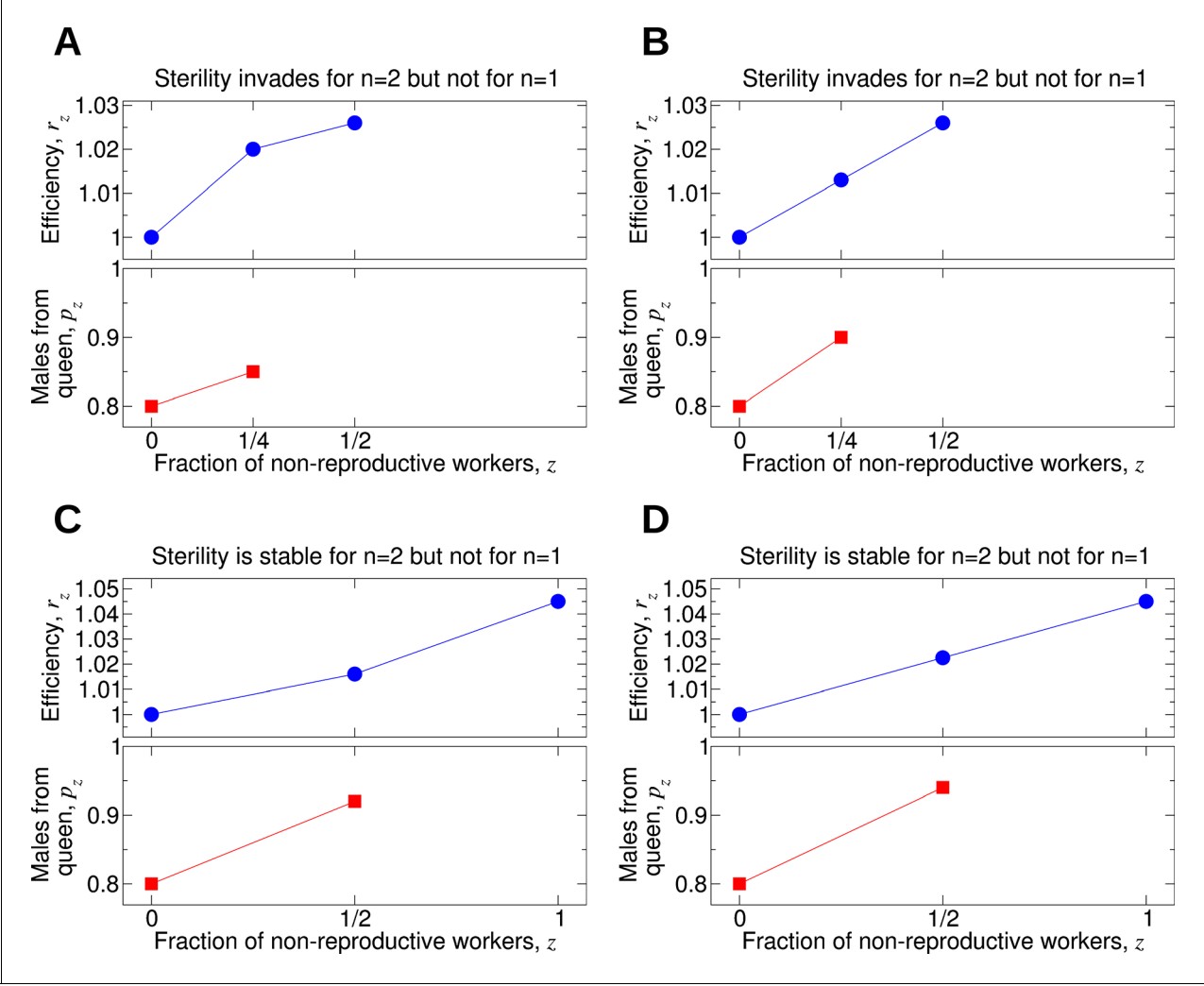

**Figure 7.** Comparing the effect of single mating ($n = 1$) and double mating ($n = 2$) on the evolution of worker sterility. Whether or not single or double mating favors the evolution of worker sterility depends on the functions $p_z$ and $r_z$. The function $p_z$ specifies the fraction of male offspring that come from the queen if a fraction, $z$, of all workers in the colony is non-reproductive. The function $r_z$ specifies the reproductive rate (or efficiency) of the colony if a fraction, $z$, of all workers in the colony is non-reproductive. We consider a recessive mutant allele, $a$, for worker sterility. (**A**, **B**) For these parameter choices, the mutant allele causing worker sterility can invade for double mating but not for single mating. (**C**, **D**) For these parameter choices, the mutant allele causing worker sterility is evolutionarily stable for double mating but not for single mating.

Hymenoptera are haplodiploid). It would be absurd to consider this to be evidence for the theoretical claim that haplodiploidy is key to the evolution of eusociality.

An important aspect of the evolution of advanced eusociality is the cancellation of all worker reproduction—in particular, the production of males. Here, we have performed a rigorous mathematical analysis of the conditions under which worker non-reproduction evolves. Our analysis has revealed that monandry does not play a crucial role in the evolution of non-reproductive worker castes. Indeed, in some cases, non-reproduction evolves when queens are multiply mated, but not when they are singly mated. Our results therefore show that the dominant paradigm, that monandry is crucial for the evolution of eusociality because it maximizes relatedness among siblings, needs to be revised. It may still turn out that monandry is important in the evolution of non-reproductive castes, but this would have to be for other reasons (**Nowak et al., 2010**; **Nonacs, 2011**; **Hunt, 2012**; **Wilson and Nowak, 2014**).

These insights have been achieved because of a more general treatment of the colony-level effects of non-reproductive workers than is allowed for by simple relatedness-based arguments. In

our model, we have explicitly accounted for two key parameters that are often neglected in other studies: $r_z$, the reproductive rate of the colony, and $p_z$, the proportion of male offspring that come from the queen, if a fraction $z$ of the colony's workers are non-reproductive. The conditions under which selection favors non-reproductive workers have been shown to depend crucially, and in interesting ways, on these two parameters. Empirical measurement of these parameters is therefore required to understand the selective forces underlying the evolution of non-reproductive castes in social insects. This suggests a line of future research.

It is important to distinguish between worker non-reproduction (workers produce no offspring, but may still retain the ability to do so), and the more specific phenomenon of worker sterility (workers do not have the ability to reproduce). An embedded distinction is that, in very many eusocial insect species, worker females have lost the ability to lay fertilized (female) eggs—e.g., through loss or degradation of the spermatheca—but retain the ability to lay unfertilized (male) eggs—i.e., they have functional ovaries (*Bourke, 1988*; *Hölldobler and Wilson, 1990*). In comparison, complete sterility (sexual and asexual) is rare in the eusocial insects; for example, only 9 of the roughly 300 genera of ants are known to have evolved complete worker sterility (*Bourke, 1995*).

In our model, we have assumed that workers can lay only unfertilized male eggs. Our model therefore best applies to those transitions from species whose workers lay only male eggs to species where the workers are non-reproductive. This is probably the most common and important route to a non-reproductive worker caste in the social insects (*Bourke, 1988*).

It is important to realize that our model applies significantly more generally than to just the (comparatively few) transitions to complete worker sterility. We have focused here on the case where a single (recessive or dominant) allele turns off worker production of males. However, mathematically, this assumption can easily be relaxed by supposing that the allele in question merely alters the frequency of worker male production by some amount. Thus, our approach is flexible enough to handle a variety of molecular mechanisms for worker reproductive restraint, which are still being elucidated for particular species (*Abouheif and Wray, 2002*; *Dearden, 2006*; *Khila and Abouheif, 2008*; *Moczek et al., 2011*; *Cameron et al., 2013*; *Sadd et al., 2015*; *Kapheim et al., 2015*).

One potentially important factor in the evolution of worker non-reproduction is worker policing. Here, if a worker lays a (male) egg, there is some chance that another worker will destroy the egg. This reduces the incentive for workers to lay male eggs in the first place, therefore selecting for decreased worker reproduction (*Bourke, 1999*; *Wenseleers et al., 2004*; *Ratnieks et al., 2006*). A slightly modified version of our model can cover the case of worker policing (with appropriate interpretations of the parameters $r_z$ and $p_z$). A detailed investigation of this situation is desirable, and is in progress. Moreover, our analysis demonstrates that worker policing, though perhaps conducive to the evolution of worker non-reproduction, is not necessary for it.

Our analysis makes no use of inclusive fitness theory, which is an unnecessary construct (*Nowak et al., 2010*; *Allen et al., 2013*). Indeed, our analysis shows that the evolution of non-reproductive workers depends on precise functional relationships between worker reproduction, queen reproduction, and colony efficiency, which inclusive fitness heuristics cannot account for. We note that inclusive fitness theory, which has dominated this area for decades, has not produced a mathematical analysis of even the most basic factors leading to the evolution of non-reproductive workers. A clear understanding of how natural selection acts on the evolution of any social behavior is possible once the field has recognized the limitations of inclusive fitness and has moved beyond them.

## Materials and methods

In this Materials and methods section, we present a mathematical model for the population dynamics of Hymenopteran colonies, and we calculate exact conditions that must be satisfied for non-reproductive workers to evolve. Our model uses haplodiploid genetics. Each female carries homologous pairs of maternal and paternal chromosomes, while each male possesses a single set of chromosomes. We assume that a specific mutation (allele) leads to non-reproductive workers. The $A$ allele represents normal behavior, while the mutant $a$ allele leads to unmated females (workers) abstaining from laying their own male eggs. If the $a$ allele is dominant, then workers that possess at least one $a$ allele are sterile. If the $a$ allele is recessive, then workers that are homozygous for $a$ are sterile. Under what conditions does the $a$ allele for non-reproductive workers invade a population?

Under what conditions is the *a* allele evolutionarily stable against invasion by *A*? A mathematical analysis of this problem lends insight into the selective forces that act on the evolution of worker sterility. While a loss of function mutation is probably recessive, it is instructive to consider both the dominant and recessive cases in detail.

## Description of the model

Consider a large population of insects. There are many colonies in the population, and each colony produces many offspring over its lifetime. The particular species under investigation has a haplodiploid genetic system. Females carry homologous pairs of maternal and paternal chromosomes, while males carry a single set of chromosomes. Queens can produce diploid female workers and gynes (future queens) from her own genotype combined with the genotype of each of the male drones that she has mated with. Queens can also produce drones using her own genotype. Female workers can produce drones as well. Thus there can be competition over whether the queen or the workers produce most of the males in a colony.

To investigate the selective forces behind non-reproductive workers—i.e., workers that do not parthenogenetically produce haploid drones—we propose two alleles, *A* and *a*. The phenotype corresponding to the *A* allele is such that workers produce drones. The phenotype corresponding to the *a* allele is such that workers do not produce drones.

An important parameter is the number, *n*, of males with which the colony's queen has mated. A schematic of the mating events is shown in **Figure 1A**. There are several possibilities: A type *AA* gyne mates with $n - m$ type *A* males and *m* type *a* males. A type *Aa* gyne mates with $n - m$ type *A* males and *m* type *a* males. A type *aa* gyne mates with $n - m$ type *A* males and *m* type *a* males.

The mating events are random. A virgin queen mates with *n* randomly chosen males in the population. Notice that, for mating, the gynes and drones are considered well-mixed: A gyne from one colony can mate with *n* drones, each chosen randomly from among the colonies in the population. For *n* = 1, we obtain single mating (monandry). For *n* = 2, we obtain double mating.

The following system of ordinary differential equations describes the selection dynamics in continuous time:

$$\dot{X}_{AA,m} = \frac{dX_{AA,m}}{dt} = \binom{n}{m} x_{AA} y_A^{n-m} y_a^m - \phi X_{AA,m}$$

$$\dot{X}_{Aa,m} = \frac{dX_{Aa,m}}{dt} = \binom{n}{m} x_{Aa} y_A^{n-m} y_a^m - \phi X_{Aa,m}$$

$$\dot{X}_{aa,m} = \frac{dX_{aa,m}}{dt} = \binom{n}{m} x_{aa} y_A^{n-m} y_a^m - \phi X_{aa,m}$$

(5)

The overdot denotes the time derivative, *d/dt*. We use the overdot notation for any time derivative.

We understand **Equation 5** as follows. We represent the genotype of a colony by the genotype of its queen and the sperm she has stored from her matings. Each queen carries homologous pairs of maternal and paternal chromosomes, so each queen has one of three possible combinations of the *A* and *a* alleles in her own genotype: *AA*, *Aa*, or *aa*. A particular queen has mated with $n - m$ type *A* males and *m* type *a* males. The number of colonies that are headed by type *AA* queens who have mated with $n - m$ type *A* males and *m* type *a* males is denoted by $X_{AA,m}$. The number of colonies that are headed by type *Aa* queens who have mated with $n - m$ type *A* males and *m* type *a* males is denoted by $X_{Aa,m}$. The number of colonies that are headed by type *aa* queens who have mated with $n - m$ type *A* males and *m* type *a* males is denoted by $X_{aa,m}$. The variables $x_{AA}$, $x_{Aa}$, and $x_{aa}$ denote the numbers of gynes of the three possible genotypes in the population. The variables $y_A$ and $y_a$ denote the numbers of drones of the two possible genotypes in the population. A gyne randomly mates with *n* drones to become a queen. The binomial coefficient accounts for all possible sequences in which a female can mate with *m* males carrying an *a* allele out of *n* total matings.

We require that the total number of colonies sums to a constant value, *c*, at all times:

$$\sum_{m=0}^{n}(X_{AA,m}+X_{Aa,m}+X_{aa,m})=c \tag{6}$$

Colonies compete for resources which are limited. Notice that $\phi$ in *Equation 5* represents a density-dependent death rate. We use $\phi$ to model the effect of environmental constraints in limiting the total number of colonies. To enforce the density constraint, *Equation 6*, on the colony variables, we set

$$\phi=c^{-1}(x_{AA}+x_{Aa}+x_{aa})(y_A+y_a)^n \tag{7}$$

Our choice to analyze the evolutionary dynamics of sterile workers in continuous time is a matter of preference. Working in continuous time usually simplifies the analysis. For example, when we derive conditions for the invasion of a recessive allele or the stability of a dominant allele, the perturbative expansion of the colony variables must be performed to second order, and the calculations become quite messy.

There are a couple of key biological parameters in our model. The emergence of sterile workers can affect the fraction of male eggs in a colony that originate from the queen. If a fraction $z$ of workers in a colony are non-reproductive, then the fraction of male offspring that originate from the queen is denoted by $p_z$. The queen and the unfertilized females may compete for production of male eggs. The function $p_z$ for $0 \leq z < 1$ likely varies for different species. It is reasonable to expect that $p_z$ is an increasing function of $z$; an increase in the proportion of workers that are non-reproductive results in a larger proportion of queen-produced males. If all workers are non-reproductive, then $z = 1$ and $p_1 = 1$.

The other key function in our model is the efficiency, $r_z$, of a colony in which a fraction $z$ of workers are non-reproductive. An appropriate biological intuition is that the parameter $r_z$ is the total number of offspring produced by a colony when a fraction, $z$, of workers in the colony are non-reproductive. As we shall see, the *ratios* of colony efficiency values, $r_z$, for colonies with different genotypes—i.e., the *relative* reproductive efficiencies of colonies with different genotypes—are important quantities for understanding the evolutionary dynamics of a mutation that causes workers to be non-reproductive. As baseline, we set $r_0 = 1$.

Non-reproductive workers forego their own reproductive potential in order to help raise their nestmates' offspring. If this division of labor has some advantage for the colony, then we expect $r_z > 1$ for some values of $z$. It is not necessary, however, that $r_z$ is a monotonically increasing function.

Since we are focused on the evolutionary dynamics of the colony variables, $X_{AA,m}$, $X_{Aa,m}$, and $X_{aa,m}$ for $0 \leq m \leq n$, we rewrite the first term on the right-hand side of *Equation 5* in terms of the colony variables. We express each of the gyne and drone numbers, $x_{AA}$, $x_{Aa}$, $x_{aa}$, $y_A$, and $y_a$, as a linear combination of the colony variables, $X_{AA,m}$, $X_{Aa,m}$, and $X_{aa,m}$. The coefficients in these linear relationships depend on whether the allele, $a$, that acts in a worker to induce that worker's sterility is dominant or recessive.

## Reproductives with a dominant sterility allele

For a dominant allele, $a$, causing worker sterility, we have the reproduction events shown in *Figure 1C*. These can be understood as follows.

Consider the offspring of type $AA,m$ colonies. The queen produces $n - m$ type $AA$ females for every $m$ type $Aa$ females. Because the queen only carries the $A$ allele, she can only produce type $A$ drones. Only workers that carry two copies of the $A$ allele are capable of making drones, and they can therefore only pass on copies of the $A$ allele, so workers are also only capable of making type $A$ drones.

Consider the offspring of type $Aa,m$ colonies. The queen produces $n - m$ type $AA$ females, $n$ type $Aa$ females, and $m$ type $aa$ females out of every $2n$ females that she produces. Because the queen carries the $A$ and $a$ alleles, she produces type $A$ and type $a$ drones in equal proportion. Only workers that carry two copies of the $A$ allele are capable of making drones, and they therefore only pass on copies of the $A$ allele, so workers are only capable of making type $A$ drones.

Consider the offspring of type $aa,m$ colonies. The queen produces $n - m$ type $Aa$ females for every $m$ type $aa$ females. Because the queen only carries the $a$ allele, she can only produce type $a$

drones. Only workers that carry two copies of the *A* allele are capable of making drones, but type *AA* workers are not produced by the queen, so workers do not produce males.

For studying the invasion of the mutant allele, only a subset of those colony types are relevant. Also, for simplicity, we neglect stochastic effects. The number of individuals produced by a colony is assumed to be very large, so that the fractions of colony offspring of the various possible genotypes are always exactly the same for that type of colony.

Our attention is on the evolution of the $3(n + 1)$ colony variables. Therefore, it is helpful to write all quantities in terms of the colony variables. We can write each type of reproductive of a colony ($x_{AA}$, $x_{Aa}$, $x_{aa}$, $y_A$, and $y_a$) as a simple weighted sum of colony variables. Using *Figure 1C*, the numbers of unfertilized females ($x_{AA}$, $x_{Aa}$, and $x_{aa}$) and males ($y_A$ and $y_a$) in the population which are capable of mating can be written as:

$$x_{AA} = \sum_{m=0}^{n} \left[ \frac{n-m}{n} g r_{\frac{m}{n}} X_{AA,m} + \frac{n-m}{2n} g r_{\frac{m+n}{2n}} X_{Aa,m} \right]$$

$$x_{Aa} = \sum_{m=0}^{n} \left[ \frac{m}{n} g r_{\frac{m}{n}} X_{AA,m} + \frac{1}{2} g r_{\frac{m+n}{2n}} X_{Aa,m} + \frac{n-m}{n} g r_1 X_{aa,m} \right]$$

$$x_{aa} = \sum_{m=0}^{n} \left[ \frac{m}{2n} g r_{\frac{m+n}{2n}} X_{Aa,m} + \frac{m}{n} g r_1 X_{aa,m} \right] \tag{8}$$

$$y_A = \sum_{m=0}^{n} \left[ k r_{\frac{m}{n}} X_{AA,m} + \frac{2 - p_{\frac{m+n}{2n}}}{2} k r_{\frac{m+n}{2n}} X_{Aa,m} \right]$$

$$y_a = \sum_{m=0}^{n} \left[ \frac{1}{2} p_{\frac{m+n}{2n}} k r_{\frac{m+n}{2n}} X_{Aa,m} + k r_1 X_{aa,m} \right]$$

Here, $0 < g \le 1$ is the fraction of all females produced by a colony that are gynes. Moreover, $0 < k \le 1$ is the fraction of all males produced by a colony that are able to mate. For example, if only a small percentage of female and male offspring of a colony eventually disperse and mate, then we have $g \ll 1$ and $k \ll 1$. The parameters $g$ and $k$ are written explicitly here for conceptual clarity. As we shall see, they turn out to be irrelevant in the conditions for invasion and stability of non-reproductive workers.

## Reproductives with a recessive sterility allele

For a recessive allele, *a*, causing worker sterility, we have the reproduction events shown in *Figure 1B*. These can be understood as follows.

Consider the offspring of type *AA,m* colonies. The queen produces $n - m$ type *AA* females for every $m$ type *Aa* females. Because the queen only carries the *A* allele, she can only produce type *A* drones. A fraction $(n - m)/n$ of all workers produce only type *A* males, and a fraction $m/n$ of all workers produce type *A* and type *a* males in equal proportion. Altogether, workers produce a fraction $(2n - m)/(2n)$ type *A* males and a fraction $m/(2n)$ type *a* males.

Consider the offspring of type *Aa,m* colonies. The queen produces $n - m$ type *AA* females, $n$ type *Aa* females, and $m$ type *aa* females out of every $2n$ females that she produces. Because the queen carries the *A* and *a* alleles, she produces type *A* and type *a* drones in equal proportion. A fraction $(n - m)/(2n)$ of all workers produce only type *A* males, and a fraction $1/2$ of all workers produce type *A* and type *a* males in equal proportion. Workers that carry two copies of the *a* allele are

non-reproductive. Altogether, workers produce a fraction $(3n - 2m)/(4n - 2m)$ type $A$ males and a fraction $n/(4n - 2m)$ type $a$ males.

Consider the offspring of type $aa,m$ colonies. The queen produces $n - m$ type $Aa$ females for every $m$ type $aa$ females. Because the queen only carries the $a$ allele, she can only produce type $a$ drones. A fraction $(n - m)/n$ of all workers produce type $A$ and type $a$ males in equal proportion. Workers that carry two copies of the $a$ allele are non-reproductive. Altogether, workers produce type $A$ and type $a$ males in equal proportion.

For studying the invasion of the mutant allele, only a subset of those colony types are relevant. Also, for simplicity, we neglect stochastic effects. The number of individuals produced by a colony is assumed to be very large, so that the fractions of colony offspring of the various possible genotypes are always exactly the same for that type of colony.

Our attention is on the evolution of the $3(n + 1)$ colony variables. Therefore, it is helpful to write all quantities in terms of the colony variables. We can write each type of reproductive of a colony ($x_{AA}$, $x_{Aa}$, $x_{aa}$, $y_A$, and $y_a$) as a simple weighted sum of colony variables. Using *Figure 1B*, the numbers of unfertilized females ($x_{AA}$, $x_{Aa}$, and $x_{aa}$) and males ($y_A$ and $y_a$) in the population which are capable of mating can be written as:

$$x_{AA} = \sum_{m=0}^{n} \left[ \frac{n-m}{n} g r_0 X_{AA,m} + \frac{n-m}{2n} g r \frac{m}{2n} X_{Aa,m} \right]$$

$$x_{Aa} = \sum_{m=0}^{n} \left[ \frac{m}{n} g r_0 X_{AA,m} + \frac{1}{2} g r \frac{m}{2n} X_{Aa,m} + \frac{n-m}{n} g r \frac{m}{n} X_{aa,m} \right]$$

$$x_{aa} = \sum_{m=0}^{n} \left[ \frac{m}{2n} g r \frac{m}{2n} X_{Aa,m} + \frac{m}{n} g r \frac{m}{n} X_{aa,m} \right]$$

$$y_A = \sum_{m=0}^{n} \left[ \frac{2n-m+mp_0}{2n} k r_0 X_{AA,m} + \frac{n\left(3 - p\frac{m}{2n}\right) - m\left(2 - p\frac{m}{2n}\right)}{2(2n-m)} k r \frac{m}{2n} X_{Aa,m} + \frac{1}{2}\left(1 - p\frac{m}{n}\right) k r \frac{m}{n} X_{aa,m} \right]$$

$$y_a = \sum_{m=0}^{n} \left[ \frac{m}{2n}(1 - p_0) k r_0 X_{AA,m} + \frac{n\left(1 + p\frac{m}{2n}\right) - mp\frac{m}{2n}}{2(2n-m)} k r \frac{m}{2n} X_{Aa,m} + \frac{1}{2}\left(1 + p\frac{m}{n}\right) k r \frac{m}{n} X_{aa,m} \right]$$

$$(9)$$

Here, $0 < g \le 1$ is the fraction of all females produced by a colony that are gynes. Moreover, $0 < k \le 1$ is the fraction of all males produced by a colony that are able to mate. For example, if only a small percentage of female and male offspring of a colony eventually disperse and mate, then we have $g \ll 1$ and $k \ll 1$. The parameters $g$ and $k$ are written explicitly here for conceptual clarity. As we shall see, they turn out to be irrelevant in the conditions for invasion and stability of non-reproductive workers.

## Rescaling of the model variables

We have described the biological intuition for our model of population genetics. To calculate conditions for understanding the evolutionary dynamics of a mutation that effects worker sterility, it is mathematically convenient to make the following substitutions:

$$
\begin{aligned}
X_{AA,m} &\longrightarrow c X_{AA,m} \\
X_{Aa,m} &\longrightarrow c X_{Aa,m} \\
X_{aa,m} &\longrightarrow c X_{aa,m} \\
x_{AA} &\longrightarrow g c x_{AA} \\
x_{Aa} &\longrightarrow g c x_{Aa} \\
x_{aa} &\longrightarrow g c x_{aa} \\
y_{A} &\longrightarrow k c y_{A} \\
y_{a} &\longrightarrow k c y_{a} \\
\phi &\longrightarrow g k^{n} c^{n} \phi \\
t &\longrightarrow g^{-1} k^{-n} c^{-n} t
\end{aligned}
\tag{10}
$$

Let's see what happens when we rescale the model variables and parameters according to *Equation 10*. We substitute *Equation 10* into *Equation 5* to obtain

$$
\begin{aligned}
\dot{X}_{AA,m} &= \frac{dX_{AA,m}}{dt} = \binom{n}{m} x_{AA} y_{A}^{n-m} y_{a}^{m} - \phi X_{AA,m} \\
\dot{X}_{Aa,m} &= \frac{dX_{Aa,m}}{dt} = \binom{n}{m} x_{Aa} y_{A}^{n-m} y_{a}^{m} - \phi X_{Aa,m} \\
\dot{X}_{aa,m} &= \frac{dX_{aa,m}}{dt} = \binom{n}{m} x_{aa} y_{A}^{n-m} y_{a}^{m} - \phi X_{aa,m}
\end{aligned}
\tag{11}
$$

We substitute *Equation 10* into *Equation 6* to obtain

$$
\sum_{m=0}^{n} (X_{AA,m} + X_{Aa,m} + X_{aa,m}) = 1
\tag{12}
$$

We substitute *Equation 10* into *Equation 7* to obtain

$$
\phi = (x_{AA} + x_{Aa} + x_{aa})(y_{A} + y_{a})^{n}
\tag{13}
$$

## Reproductives (rescaled) with a dominant sterility allele

We substitute *Equation 10* into *Equation 8* to obtain

$$x_{AA} = \sum_{m=0}^{n} \left[ \frac{n-m}{n} r_{\frac{m}{n}} X_{AA,m} + \frac{n-m}{2n} r_{\frac{m+n}{2n}} X_{Aa,m} \right]$$

$$x_{Aa} = \sum_{m=0}^{n} \left[ \frac{m}{n} r_{\frac{m}{n}} X_{AA,m} + \frac{1}{2} r_{\frac{m+n}{2n}} X_{Aa,m} + \frac{n-m}{n} r_1 X_{aa,m} \right]$$

$$x_{aa} = \sum_{m=0}^{n} \left[ \frac{m}{2n} r_{\frac{m+n}{2n}} X_{Aa,m} + \frac{m}{n} r_1 X_{aa,m} \right] \tag{14}$$

$$y_A = \sum_{m=0}^{n} \left[ r_{\frac{m}{n}} X_{AA,m} + \frac{2 - p_{\frac{m+n}{2n}}}{2} r_{\frac{m+n}{2n}} X_{Aa,m} \right]$$

$$y_a = \sum_{m=0}^{n} \left[ \frac{1}{2} p_{\frac{m+n}{2n}} r_{\frac{m+n}{2n}} X_{Aa,m} + r_1 X_{aa,m} \right]$$

## Reproductives (rescaled) with a recessive sterility allele

We substitute *Equation 10* into *Equation 9* to obtain

$$x_{AA} = \sum_{m=0}^{n} \left[ \frac{n-m}{n} r_0 X_{AA,m} + \frac{n-m}{2n} r_{\frac{m}{2n}} X_{Aa,m} \right]$$

$$x_{Aa} = \sum_{m=0}^{n} \left[ \frac{m}{n} r_0 X_{AA,m} + \frac{1}{2} r_{\frac{m}{2n}} X_{Aa,m} + \frac{n-m}{n} r_{\frac{m}{n}} X_{aa,m} \right]$$

$$x_{aa} = \sum_{m=0}^{n} \left[ \frac{m}{2n} r_{\frac{m}{2n}} X_{Aa,m} + \frac{m}{n} r_{\frac{m}{n}} X_{aa,m} \right]$$

$$y_A = \sum_{m=0}^{n} \left[ \frac{2n-m+mp_0}{2n} r_0 X_{AA,m} + \frac{n\left(3 - p_{\frac{m}{2n}}\right) - m\left(2 - p_{\frac{m}{2n}}\right)}{2(2n-m)} r_{\frac{m}{2n}} X_{Aa,m} + \frac{1}{2}\left(1 - p_{\frac{m}{n}}\right) r_{\frac{m}{n}} X_{aa,m} \right]$$

$$y_a = \sum_{m=0}^{n} \left[ \frac{m}{2n}(1 - p_0) r_0 X_{AA,m} + \frac{n\left(1 + p_{\frac{m}{2n}}\right) - mp_{\frac{m}{2n}}}{2(2n-m)} r_{\frac{m}{2n}} X_{Aa,m} + \frac{1}{2}\left(1 + p_{\frac{m}{n}}\right) r_{\frac{m}{n}} X_{aa,m} \right]$$

$$\tag{15}$$

## Conditions for evolutionary invasion and evolutionary stability of worker sterility: perturbative analysis

Notice that when we rescale the model variables and parameters according to *Equation 10*, the evolutionary dynamics are mathematically unchanged: *Equation 5* has the same form as *Equation 11*, *Equation 6* has the same form as *Equation 12*, *Equation 7* has the same form as *Equation 13*, *Equation 8* has the same form as *Equation 14*, and *Equation 9* has the same form as *Equation 15*. But it is apparent why the rescalings (*Equation 10*) are helpful in doing calculations: When the right-hand side of (*Equation 11*) is written out in terms of the colony frequency variables $X_{AA,m}$, $X_{Aa,m}$, and $X_{aa,m}$, the parameters $g$, $k$, and $c$, which are not essential for understanding the evolutionary invasion or evolutionary stability of non-reproductive workers, no longer appear in the calculations. This simplifies writing and improves clarity in the calculations that follow.

To begin, note that only two pure equilibria are possible:

- $X_{AA,0} = 1$ with all other $X$'s equal to zero. In this case, the $a$ allele does not exist in any individual in the population.

- $X_{aa,n} = 1$ with all other $X$'s equal to zero. In this case, the $A$ allele does not exist in any individual in the population.

From *Equation 11*, if any mixed equilibria exist, then they will feature $3(n + 1)$ nonzero frequencies.

## Invasion of a dominant worker sterility allele

What happens if we start with an infinitesimal quantity of the mutant allele, $a$, by perturbing the $X_{AA,0} = 1$ pure equilibrium: $X_{AA,0} \longrightarrow 1 - \epsilon \delta_{AA,0}^{(1)}$, with $\epsilon \ll 1$? Does a dominant worker sterility allele spread in the population, or is it eliminated?

Although the state space is $(3n + 2)$-dimensional ($3n + 3$ types of colonies subject to the density constraint), the analysis simplifies. Provided that the perturbation is small (i.e. that $\epsilon \ll 1$), only three colony types, *AA*,0, *AA*,1, and *Aa*,0, determine whether or not the dominant worker sterility allele invades. Any other colony type is headed by a queen that possesses at least two mutant $a$ alleles (from her own genotype combined with the sperm she has collected), but such queens are so rare as to be negligible. The relevant equations among (*Equation 11*) for studying invasion of a dominant sterility allele are

$$
\begin{aligned}
\dot{X}_{AA,0} &= x_{AA} y_A^n - \phi X_{AA,0} \\
\dot{X}_{AA,1} &= n x_{AA} y_A^{n-1} y_a - \phi X_{AA,1} \\
\dot{X}_{Aa,0} &= x_{Aa} y_A^n - \phi X_{Aa,0}
\end{aligned}
\tag{16}
$$

Formally keeping track of powers of $\epsilon$, and disregarding higher-order terms, we have:

$$
\begin{aligned}
X_{AA,0} &= 1 - \epsilon \delta_{AA,0}^{(1)} - \mathcal{O}(\epsilon^2) \\
X_{AA,1} &= +\epsilon \delta_{AA,1}^{(1)} + \mathcal{O}(\epsilon^2) \\
X_{Aa,0} &= +\epsilon \delta_{Aa,0}^{(1)} + \mathcal{O}(\epsilon^2)
\end{aligned}
\tag{17}
$$

To simplify the density constraint (*Equation 12*) for our calculation, we substitute (*Equation 17*) into (*Equation 12*) and collect powers of $\epsilon$. We get

$$
\delta_{AA,0}^{(1)} = \delta_{AA,1}^{(1)} + \delta_{Aa,0}^{(1)}
\tag{18}
$$

Next, we substitute (*Equation 17*) into (*Equation 14*), using the density constraint (*Equation 18*) and keeping terms only up to order $\epsilon$:

$$x_{AA} = r_0 + \epsilon \left[ \frac{\dfrac{(n-1)r_{\frac{1}{n}} - nr_0}{n}}{n} \delta_{AA,1}^{(1)} + \frac{-2r_0 + r_{\frac{1}{2}}}{2} \delta_{Aa,0}^{(1)} \right] + \mathcal{O}(\epsilon^2)$$

$$x_{Aa} = \epsilon \left[ \frac{r_{\frac{1}{n}}}{n} \delta_{AA,1}^{(1)} + \frac{r_{\frac{1}{2}}}{2} \delta_{Aa,0}^{(1)} \right] + \mathcal{O}(\varepsilon^2)$$

$$y_A = r_0 + \varepsilon \left[ -\left( r_0 - r_{\frac{1}{n}} \right) \delta_{AA,1}^{(1)} + \frac{\left( \dfrac{2-p_{\frac{1}{2}}}{2} \right) r_{\frac{1}{2}} - 2r_0}{2} \delta_{Aa,0}^{(1)} \right] + \mathcal{O}(\varepsilon^2) \qquad (19)$$

$$y_a = \varepsilon \left[ \frac{p_{\frac{1}{2}} r_{\frac{1}{2}}}{2} \delta_{Aa,0}^{(1)} \right] + \mathcal{O}(\varepsilon^2)$$

By plugging (*Equation 19*) and (*Equation 17*) into (*Equation 16*), using the density constraint (*Equation 18*), and collecting powers of $\epsilon$, we find

$$\dot{\delta}_{AA,1}^{(1)} = -r_0^{n+1} \delta_{AA,1}^{(1)} + \frac{\dfrac{n p_{\frac{1}{2}} r_{\frac{1}{2}} r_0^n}{2}}{2} \delta_{Aa,0}^{(1)}$$

$$\dot{\delta}_{Aa,0}^{(1)} = \frac{\dfrac{r_{\frac{1}{n}} r_0^n}{n}}{n} \delta_{AA,1}^{(1)} + \frac{-2r_0^{n+1} + r_{\frac{1}{2}} r_0^n}{2} \delta_{Aa,0}^{(1)}$$

The equations for $\dot{\delta}_{AA,1}^{(1)}$ and $\dot{\delta}_{Aa,0}^{(1)}$ can be written in matrix form as

$$\begin{pmatrix} \dot{\delta}_{AA,1}^{(1)} \\ \dot{\delta}_{Aa,0}^{(1)} \end{pmatrix} = \begin{pmatrix} -r_0^{n+1} & \dfrac{\dfrac{n p_{\frac{1}{2}} r_{\frac{1}{2}} r_0^n}{2}}{2} \\ \dfrac{r_{\frac{1}{n}} r_0^n}{n} & \dfrac{-2r_0^{n+1} + r_{\frac{1}{2}} r_0^n}{2} \end{pmatrix} \begin{pmatrix} \delta_{AA,1}^{(1)} \\ \delta_{Aa,0}^{(1)} \end{pmatrix}$$

Setting the dominant eigenvalue to be greater than zero and simplifying, we find that the dominant allele for worker sterility increases in frequency if

$$\frac{r_{\frac{1}{2}}}{r_0} \left[ 1 + p_{\frac{1}{2}} \left( \frac{r_{\frac{1}{n}}}{r_0} \right) \right] > 2 \qquad (20)$$

Depending on the values of the parameters $p_{1/2}$, $r_0$, $r_{1/2}$, and $r_{1/n}$, non-reproductive workers may or may not evolve with any number of matings, $n$, of the queen. In *Figure 8*, we show the regions of parameter space for which non-reproductive workers can or cannot evolve for a dominant sterility allele. Holding all other parameters constant, an increase in $r_1$ favors the invasion of the dominant sterility allele for $n = 1$. This is easy to see in *Figure 8*: the upper panels (higher $r_1$) correspond to invasion of the dominant sterility allele for $n = 1$, while the lower panels do not. Holding all other

parameters constant, an increase in $r_{1/2}$ favors the invasion of the dominant sterility allele for $n = 2$. Again, this is seen in **Figure 8**: the right panels (higher $r_{1/2}$) represent invasion of the dominant sterility allele for $n = 2$, while the left panels do not. Additionally, holding all other parameters constant, an increase in $r_{1/2}$ facilitates the invasion of the dominant sterility allele for $n = 1$. This can also be seen in **Figure 8**: the boundary separating the upper panels from the lower panels decreases as $r_{1/2}$ increases. If $r_{1/2} > r_1$, then it is possible that a dominant sterility allele invades for double mating but not for single mating.

## Invasion of a recessive worker sterility allele

What happens if we start with an infinitesimal quantity of the mutant allele, $a$, by perturbing the $X_{AA,0} = 1$ pure equilibrium: $X_{AA,0} \longrightarrow 1 - \epsilon \delta_{AA,0}^{(1)}$, with $\epsilon \ll 1$? Does a recessive worker sterility allele spread in the population, or is it eliminated?

Although the state space is $(3n + 2)$-dimensional ($3n + 3$ types of colonies subject to the density constraint), the analysis again simplifies. Provided that the perturbation is small (i.e. that $\epsilon \ll 1$), only six colony types, $AA,0$, $AA,1$, $Aa,0$, $AA,2$, $Aa,1$, and $aa,0$, determine whether or not the recessive worker sterility allele invades. Any other colony type is headed by a queen that possesses at least three mutant $a$ alleles (from her own genotype combined with the sperm she has collected), but such queens are so rare as to be negligible. The relevant equations among (**Equation 11**) for studying invasion of a recessive sterility allele are

$$
\begin{aligned}
\dot{X}_{AA,0} &= x_{AA} y_A^n - \phi X_{AA,0} \\
\dot{X}_{AA,1} &= n x_{AA} y_A^{n-1} y_a - \phi X_{AA,1} \\
\dot{X}_{Aa,0} &= x_{Aa} y_A^n - \phi X_{Aa,0} \\
\dot{X}_{AA,2} &= \frac{n(n-1)}{2} x_{AA} y_A^{n-2} y_a^2 - \phi X_{AA,2} \\
\dot{X}_{Aa,1} &= n x_{Aa} y_A^{n-1} y_a - \phi X_{Aa,1} \\
\dot{X}_{aa,0} &= x_{aa} y_A^n - \phi X_{aa,0}
\end{aligned}
\tag{21}
$$

Recall that for analysis of the dominant allele, we only needed to consider terms of order $\epsilon$ to derive conditions for invasion of the allele. For analysis of the recessive allele, terms of order $\epsilon$ do not provide

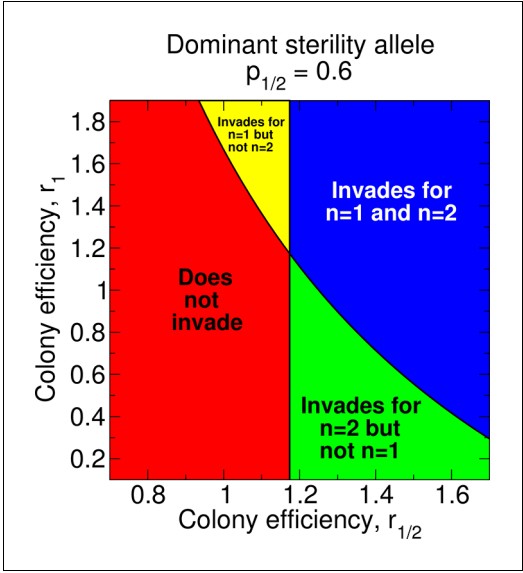

**Figure 8.** Regions of the parameter space and evolution of worker sterility for a dominant sterility allele. The evolutionary invasion of a dominant worker sterility allele depends on values of the parameters $p_{1/2}$, $r_{1/2}$, and $r_1$ for single mating, $n = 1$, and on values of the parameters $p_{1/2}$ and $r_{1/2}$ for double mating, $n = 2$. The figure shows four parameter regions indicating whether or not worker sterility can evolve for single or double mating. We set $r_0 = 1$ as baseline.

sufficient information for determining if the allele invades, which adds a level of tedium to the calculation. Formally keeping track of powers of $\epsilon$ and $\epsilon^2$, and disregarding higher-order terms, we have:

$$
\begin{aligned}
X_{AA,0} &= 1 - \epsilon \delta_{AA,0}^{(1)} - \epsilon^2 \delta_{AA,0}^{(2)} - \mathcal{O}(\epsilon^3) \\
X_{AA,1} &= +\epsilon \delta_{AA,1}^{(1)} + \epsilon^2 \delta_{AA,1}^{(2)} + \mathcal{O}(\epsilon^3) \\
X_{Aa,0} &= +\epsilon \delta_{Aa,0}^{(1)} + \epsilon^2 \delta_{Aa,0}^{(2)} + \mathcal{O}(\epsilon^3) \\
X_{AA,2} &= +\epsilon^2 \delta_{AA,2}^{(2)} + \mathcal{O}(\epsilon^3) \\
X_{Aa,1} &= +\epsilon^2 \delta_{Aa,1}^{(2)} + \mathcal{O}(\epsilon^3) \\
X_{aa,0} &= +\epsilon^2 \delta_{aa,0}^{(2)} + \mathcal{O}(\epsilon^3)
\end{aligned}
\tag{22}
$$

The simplified density constraint, *Equation 18*, holds regardless of whether the sterility allele under consideration is dominant or recessive. To further simplify the density constraint (*Equation 12*) for the case of a recessive sterility allele, we substitute (*Equation 22*) into (*Equation 12*) and collect powers of $\epsilon^2$. We get

$$
\delta_{AA,0}^{(2)} = \delta_{AA,1}^{(2)} + \delta_{Aa,0}^{(2)} + \delta_{AA,2}^{(2)} + \delta_{Aa,1}^{(2)} + \delta_{aa,0}^{(2)}
\tag{23}
$$

Next, we substitute (*Equation 22*) into (*Equation 15*), using the density constraints (*Equation 18*) and (*Equation 23*) and keeping terms up to order $\epsilon^2$:

$$
x_{AA}r_0^{-1} = 1 + \epsilon\left[\frac{-1}{n}\delta_{AA,1}^{(1)} - \frac{1}{2}\delta_{Aa,0}^{(1)}\right] + \epsilon^2\left[\frac{-1}{n}\delta_{AA,1}^{(2)} - \frac{1}{2}\delta_{Aa,0}^{(2)} - \frac{2}{n}\delta_{AA,2}^{(2)} + \frac{-2n + (n-1)r_{\frac{1}{2n}}r_0^{-1}}{2n}\delta_{Aa,1}^{(2)} - \delta_{aa,0}^{(2)}\right] + \mathcal{O}(\epsilon^3)
$$

$$
x_{Aa}r_0^{-1} = \epsilon\left[\frac{1}{n}\delta_{AA,1}^{(1)} + \frac{1}{2}\delta_{Aa,0}^{(1)}\right] + \epsilon^2\left[\frac{1}{n}\delta_{AA,1}^{(2)} + \frac{1}{2}\delta_{Aa,0}^{(2)} + \frac{2}{n}\delta_{AA,2}^{(2)} + \frac{r_{\frac{1}{2n}}r_0^{-1}}{2}\delta_{Aa,1}^{(2)} + \delta_{aa,0}^{(2)}\right] + \mathcal{O}(\epsilon^3)
$$

$$
x_{aa}r_0^{-1} = \epsilon^2\left[\frac{r_{\frac{1}{2n}}r_0^{-1}}{2n}\delta_{Aa,1}^{(2)}\right] + \mathcal{O}(\epsilon^3)
$$

$$
\begin{aligned}
y_A r_0^{-1} = {}& 1 + \epsilon\left[-\frac{1-p_0}{2n}\delta_{AA,1}^{(1)} - \frac{1+p_0}{4}\delta_{Aa,0}^{(1)}\right] + \epsilon^2\Big[-\frac{1-p_0}{2n}\delta_{AA,1}^{(2)} - \frac{1+p_0}{4}\delta_{Aa,0}^{(2)} - \frac{1-p_0}{n}\delta_{AA,2}^{(2)} \\
&+ \frac{2-4n - \left[2 - n\left(3 - p_{\frac{1}{2n}}\right) - p_{\frac{1}{2n}}\right]r_{\frac{1}{2n}}r_0^{-1}}{2(2n-1)}\delta_{Aa,1}^{(2)} - \frac{1+p_0}{2}\delta_{aa,0}^{(2)}\Big] + \mathcal{O}(\epsilon^3)
\end{aligned}
$$

$$
\begin{aligned}
y_a r_0^{-1} = {}& \epsilon\left[\frac{1-p_0}{2n}\delta_{AA,1}^{(1)} + \frac{1+p_0}{4}\delta_{Aa,0}^{(1)}\right] + \epsilon^2\Big[\frac{1-p_0}{2n}\delta_{AA,1}^{(2)} + \frac{1+p_0}{4}\delta_{Aa,0}^{(2)} + \frac{1-p_0}{n}\delta_{AA,2}^{(2)} \\
&+ \frac{\left[n + (n-1)p_{\frac{1}{2n}}\right]r_{\frac{1}{2n}}r_0^{-1}}{2(2n-1)}\delta_{Aa,1}^{(2)} + \frac{1+p_0}{2}\delta_{aa,0}^{(2)}\Big] + \mathcal{O}(\epsilon^3)
\end{aligned}
$$

$$
\tag{24}
$$

By plugging (*Equation 24*) and (*Equation 22*) into (*Equation 21*), using the density constraint (*Equation 18*), and collecting powers of $\epsilon$, we find

$$\dot{\delta}_{AA,1}^{(1)} = \frac{-(1+p_0)}{2}r_0^{n+1}\delta_{AA,1}^{(1)} + \frac{n(1+p_0)}{4}r_0^{n+1}\delta_{Aa,0}^{(1)}$$

$$\dot{\delta}_{Aa,0}^{(1)} = \frac{1}{n}r_0^{n+1}\delta_{AA,1}^{(1)} - \frac{1}{2}r_0^{n+1}\delta_{Aa,0}^{(1)}$$

The equations for $\dot{\delta}_{AA,1}^{(1)}$ and $\dot{\delta}_{Aa,0}^{(1)}$ can be written in matrix form as

$$\begin{pmatrix} \dot{\delta}_{AA,1}^{(1)} \\ \dot{\delta}_{Aa,0}^{(1)} \end{pmatrix} = r_0^{n+1} \begin{pmatrix} \frac{-(1+p_0)}{2} & \frac{n(1+p_0)}{4} \\ \frac{1}{n} & \frac{-1}{2} \end{pmatrix} \begin{pmatrix} \delta_{AA,1}^{(1)} \\ \delta_{Aa,0}^{(1)} \end{pmatrix}$$

The two eigenvectors ($v_0$ and $v_-$) and their corresponding eigenvalues ($\lambda_0$ and $\lambda_-$) are

$$v_0 = \begin{pmatrix} n \\ 2 \end{pmatrix} \quad \lambda_0 = 0$$

$$v_- = \begin{pmatrix} n(1+p_0) \\ -2 \end{pmatrix} \quad \lambda_- = \frac{-(2+p_0)}{2}r_0^{n+1}$$

Notice that the dominant eigenvalue is equal to zero, so a computation to leading order in $\epsilon$ cannot provide information on the invasion of the recessive sterility allele.

We can also see this more formally. An arbitrary initial perturbation to a resident $A$ population can be written as a linear superposition of the eigenvectors $v_0$ and $v_-$:

$$\begin{pmatrix} \delta_{AA,1}^{(1)} \\ \delta_{Aa,0}^{(1)} \end{pmatrix} = C_0 \begin{pmatrix} n \\ 2 \end{pmatrix} + C_- \begin{pmatrix} n(1+p_0) \\ -2 \end{pmatrix} \exp\left(\frac{-(2+p_0)}{2}r_0^{n+1}t\right) \tag{25}$$

Here $C_0$ and $C_-$ are constants. We can substitute (*Equation 24*) and (*Equation 22*) into (*Equation 21*), using the density constraints (*Equation 18*) and (*Equation 23*), keeping terms of order $\epsilon$ and $\epsilon^2$, and dividing each term by one factor of $\epsilon$. We obtain

$$
\begin{aligned}
\left[-\dot{\delta}_{AA,0}^{(1)} - \epsilon\dot{\delta}_{AA,0}^{(2)}\right]r_0^{-(n+1)} =\ & \frac{2-n-np_0}{4n}\left(-2\delta_{AA,1}^{(1)} + n\delta_{Aa,0}^{(1)}\right) \\
& +\epsilon\Big[\frac{2-n-np_0}{4n}\left(-2\delta_{AA,1}^{(2)} + n\delta_{Aa,0}^{(2)}\right) \\
& +\frac{-2+np_0}{n}\delta_{AA,2}^{(2)} \\
& +\frac{r_{\frac{1}{2n}}r_0^{-1} - n\left[2 + r_{\frac{1}{2n}}r_0^{-1} - n\left(4 - \left(2 + n + (n-1)p_{\frac{1}{2n}}\right)r_{\frac{1}{2n}}r_0^{-1}\right)\right]}{2n(2n-1)}\delta_{Aa,1}^{(2)} \\
& -\frac{n(1+p_0)}{2}\delta_{aa,0}^{(2)} \\
& +\frac{(1-p_0)[3+n(1-p_0)+p_0]}{8n}[\delta_{AA,1}^{(1)}]^2 \\
& +\frac{n(1+p_0)[3+n+(n-1)p_0]}{32}[\delta_{Aa,0}^{(1)}]^2 \\
& +\frac{3+n-(n-1)p_0^2}{8}\delta_{AA,1}^{(1)}\delta_{Aa,0}^{(1)}\Big]
\end{aligned}
$$

$$\tag{26}$$

We can again use the density constraints (*Equation 18*) and (*Equation 23*) to rewrite the left-hand side of (*Equation 26*). We can also substitute the general solution for the quantities $\delta^{(1)}_{AA,1}$ and $\delta^{(1)}_{Aa,0}$, *Equation 25*, into the right-hand side of (*Equation 26*):

$$
\begin{aligned}
&\left[-\dot{\delta}^{(1)}_{AA,1}-\dot{\delta}^{(1)}_{Aa,0}\right]r_0^{-(n+1)} \\
&+\epsilon[-\dot{\delta}^{(2)}_{AA,1}-\dot{\delta}^{(2)}_{Aa,0} \\
&\quad -\dot{\delta}^{(2)}_{AA,2}-\dot{\delta}^{(2)}_{Aa,1} \\
&\quad -\dot{\delta}^{(2)}_{aa,0}]r_0^{-(n+1)} \quad = \quad
\begin{aligned}
&\frac{2-n-np_o}{4n}[-2\left(nC_0+n(1+p_o)C_-\exp\left(\frac{-(2+p_0)}{2}r_0^{n+1}t\right)\right) \\
&\quad +n\left(2C_0-2C_-\exp\left(\frac{-(2+p_0)}{2}r_0^{n+1}t\right)\right)] \\
&+\epsilon[\frac{2-n-np_0}{4n}\left(-2\delta^{(2)}_{AA,1}+n\delta^{(2)}_{Aa,0}\right) \\
&\quad +\frac{-2+np_0}{n}\delta^{(2)}_{AA,2} \\
&\quad +\frac{\frac{r_1}{2n}r_0^{-1}-n[2+\frac{r_1}{2n}r_0^{-1}-n\left(4-\left(2+n+(n-1)p\frac{1}{2n}\right)\frac{r_1}{2n}r_0^{-1}\right)]}{2n(2n-1)}\delta^{(2)}_{Aa,1} \\
&\quad -\frac{n(1+p_0)}{2}\delta^{(2)}_{aa,0} \\
&\quad +\frac{(1-p_0)[3+n(1-p_0)+p_0]}{8n}[\delta^{(1)}_{AA,1}]^2 \\
&\quad +\frac{n(1+p_0)[3+n+(n-1)p_0]}{32}[\delta^{(1)}_{Aa,0}]^2 \\
&\quad +\frac{3+n-(n-1)p_0^2}{8}\delta^{(1)}_{AA,1}\delta^{(1)}_{Aa,0}]
\end{aligned}
\end{aligned}
$$

(27)

Note that each term in (*Equation 27*) involving the quantities $\delta^{(2)}_{AA,1}$, $\delta^{(2)}_{Aa,0}$, $\delta^{(2)}_{AA,2}$, $\delta^{(2)}_{Aa,1}$, and $\delta^{(2)}_{aa,0}$ is multiplied by $\epsilon$. In the limit $\epsilon \longrightarrow 0$, the quantities $\delta^{(2)}_{AA,1}$, $\delta^{(2)}_{Aa,0}$, $\delta^{(2)}_{AA,2}$, $\delta^{(2)}_{Aa,1}$, and $\delta^{(2)}_{aa,0}$ do not affect the dynamics of the quantities $\delta^{(1)}_{AA,1}$ and $\delta^{(1)}_{Aa,0}$. However, the quantities $\delta^{(1)}_{AA,1}$ and $\delta^{(1)}_{Aa,0}$ alone tell us nothing about whether or not the recessive sterility allele invades a resident $A$ population. Therefore, we must consider the terms of order $\epsilon^2$ in our dynamical equations (*Equation 21*) to determine if a rare $a$ allele can invade a resident $A$ population. In our calculations that follow, we use the eigenvector $v_0$ corresponding to the zero eigenvalue, i.e.

$$
\begin{pmatrix}\delta^{(1)}_{AA,1}\\\delta^{(1)}_{Aa,0}\end{pmatrix}=\frac{\delta^{(1)}_{AA,0}}{n+2}\begin{pmatrix}n\\2\end{pmatrix}
$$

(28)

Substituting (*Equation 24*), (*Equation 22*), and (*Equation 28*) into (*Equation 21*), using the density constraints (*Equation 18*) and (*Equation 23*), and keeping terms of order $\epsilon^2$, we have

$$-\dot{\delta}_{AA,0}^{(2)} r_0^{-(n+1)} = \frac{2-n-np_0}{4n}\left(-2\delta_{AA,1}^{(2)} - n\delta_{Aa,0}^{(2)}\right)$$

$$+\frac{-2+np_0}{n}\delta_{AA,2}^{(2)}$$

$$+\frac{r_{\frac{1}{2n}} r_0^{-1} - n\left[2 + r_{\frac{1}{2n}} r_0^{-1} - n\left(4 - \left(2+n+(n-1)p_{\frac{1}{2n}}\right) r_{\frac{1}{2n}} r_0^{-1}\right)\right]}{2n(2n-1)}\delta_{Aa,1}^{(2)} \quad (29)$$

$$-\frac{n(1+p_0)}{2}\delta_{aa,0}^{(2)}$$

$$+\frac{n(n+3)}{2(n+2)^2}[\delta_{AA,0}^{(1)}]^2$$

We also have

$$\dot{\delta}_{AA,1}^{(2)} r_0^{-(n+1)} = \frac{1+p_0}{4}\left(-2\delta_{AA,1}^{(2)} + n\delta_{Aa,0}^{(2)}\right)$$

$$+(1-p_0)\delta_{AA,2}^{(2)}$$

$$+\frac{n[n+(n-1)p_{\frac{1}{2n}}]}{2(2n-1)} r_{\frac{1}{2n}} r_0^{-1}\delta_{Aa,1}^{(2)}$$

$$+\frac{n(1+p_0)}{2}\delta_{aa,0}^{(2)}$$

$$-\frac{n(n+1)}{(n+2)^2}[\delta_{AA,0}^{(1)}]^2$$

$$\dot{\delta}_{Aa,0}^{(2)} r_0^{-(n+1)} = \frac{-1}{2n}\left(-2\delta_{AA,1}^{(2)} + n\delta_{Aa,0}^{(2)}\right)$$

$$+\frac{2}{n}\delta_{AA,2}^{(2)}$$

$$+\frac{1}{2}r_{\frac{1}{2n}} r_0^{-1}\delta_{Aa,1}^{(2)}$$

$$+\delta_{aa,0}^{(2)}$$

$$-\frac{2n}{(n+2)^2}[\delta_{AA,0}^{(1)}]^2$$

$$\dot{\delta}_{AA,2}^{(2)} r_0^{-(n+1)} = -\delta_{AA,2}^{(2)} + \frac{n(n-1)}{2(n+2)^2}[\delta_{AA,0}^{(1)}]^2$$

$$\dot{\delta}_{Aa,1}^{(2)} r_0^{-(n+1)} = -\delta_{Aa,1}^{(2)} + \frac{2n}{(n+2)^2}[\delta_{AA,0}^{(1)}]^2$$

$$\dot{\delta}_{aa,0}^{(2)} r_0^{-(n+1)} = -\delta_{aa,0}^{(2)} + \frac{1}{2n}r_{\frac{1}{2n}} r_0^{-1}\delta_{Aa,1}^{(2)}$$

We can directly integrate the equation for $\dot{\delta}_{AA,2}^{(2)}$. We get

$$\delta_{AA,2}^{(2)} = \frac{n(n-1)}{2(n+2)^2} [\delta_{AA,0}^{(1)}]^2 [1 - \exp\left(-r_0^{n+1}t\right)] \tag{30}$$

We can also directly integrate the equation for $\dot{\delta}_{Aa,1}^{(2)}$. We get

$$\delta_{Aa,1}^{(2)} = \frac{2n}{(n+2)^2} [\delta_{AA,0}^{(1)}]^2 [1 - \exp\left(-r_0^{n+1}t\right)] \tag{31}$$

We can use the solution for $\delta_{Aa,1}^{(2)}$ to solve for $\delta_{aa,0}^{(2)}$. We get

$$\delta_{aa,0}^{(2)} = \frac{r_{\frac{1}{2n}}}{r_0(n+2)^2} [\delta_{AA,0}^{(1)}]^2 [1 - (1 + r_0^{n+1}t) \exp\left(-r_0^{n+1}t\right)] \tag{32}$$

Manipulating the equations for $\dot{\delta}_{AA,1}^{(2)}$ and $\dot{\delta}_{Aa,0}^{(2)}$, we find that

$$r_0^{-(n+1)} \frac{d}{dt}(-2\delta_{AA,1}^{(2)} + n\delta_{Aa,0}^{(2)}) = \frac{-(2+p_0)}{2}\left(-2\delta_{AA,1}^{(2)} + n\delta_{Aa,0}^{(2)}\right) + 2p_0\delta_{AA,2}^{(2)}$$

$$-\frac{n[1 + 2(n-1)p_{\frac{1}{2n}}]}{2(2n-1)} r_{\frac{1}{2n}} r_0^{-1}\delta_{Aa,1}^{(2)}$$

$$-np_0\delta_{aa,0}^{(2)}$$

$$+\frac{2n}{(n+2)^2}[\delta_{AA,0}^{(1)}]^2$$

We can integrate this equation to solve for the quantity $-2\delta_{AA,1}^{(2)} + n\delta_{Aa,0}^{(2)}$. We obtain

$$-2\delta_{AA,1}^{(2)} + n\delta_{Aa,0}^{(2)} = [\frac{2n[p_0 - n\left(1 + 2p_0 + 2(n-1)p_{\frac{1}{2n}}\right)]r_{\frac{1}{2n}}r_0^{-1}}{(n+2)^2(2+p_0)(2n-1)}$$

$$+\frac{2n[2+(n-1)p_0]}{(n+2)^2(2+p_0)}][\delta_{AA,0}^{(1)}]^2$$

$$+[\frac{2n[2-3n+2(n-1)np_{\frac{1}{2n}}]r_{\frac{1}{2n}}r_0^{-1}}{(n+2)^2p_0(2n-1)}$$

$$-\frac{2n[n-1-r_{\frac{1}{2n}}r_0^{-1}(1+r_0^{n+1}t)]}{(n+2)^2}][\delta_{AA,0}^{(1)}]^2\exp\left(-r_0^{n+1}t\right) \tag{33}$$

$$+[\frac{4n[n\left(3 - 2(n-1)p_{\frac{1}{2n}}\right) - 2]r_{\frac{1}{2n}}r_0^{-1}}{(n+2)^2p_0(2+p_0)(2n-1)}$$

$$+\frac{4n(n-2)}{(n+2)^2(2+p_0)}][\delta_{AA,0}^{(1)}]^2 \exp\left(\frac{-(2+p_0)}{2}r_0^{n+1}t\right)$$

To determine if the resident *A* population is unstable to invasion by the *a* allele, we must consider the regime $1 \ll t \ll \epsilon^{-1}$. Notice that on a short time scale, each of the time-dependent terms in *Equations 30–33* will approach zero. We must consider the sign of $\dot{\delta}_{AA,0}^{(2)}$ in the limit of large times

$t \gg 1$ but before the terms in (*Equation 22*) become comparable in magnitude. Our condition for invasion of the sterility allele is therefore

$$\lim_{\substack{\epsilon t \longrightarrow 0 \\ t \longrightarrow \infty}} \dot{\delta}^{(2)}_{AA,0} > 0 \tag{34}$$

Substituting (*Equations 29–33*) into (*Equation 34*), we find that the recessive allele for worker sterility increases in frequency if

$$\frac{r_{\frac{1}{2n}}}{r_0} > \frac{2(2n-1)(2+n+np_0)}{2n^2\left(2+p_0+p_{\frac{1}{2n}}\right) + n\left(3+3p_0-2p_{\frac{1}{2n}}\right) - 2(1+p_0)}$$

In the Results, we focused on single and double mating. *Figure 3A* shows that fairly large efficiency increases from non-reproductive workers (around 10–20%) are needed for sterility to invade.

*Figure 3A* also shows how the number of matings affects the invasion of non-reproductive workers for different values of the parameters $r_{1/4}$ and $r_{1/2}$. Sample forms of the functions $p_z$ and $r_z$ are shown in *Figure 7A,B*. For *Figure 7A*, we have $p_0 = 0.8$, $p_{1/4} = 0.85$, $r_0 = 1$, $r_{1/4} = 1.02$, and $r_{1/2} = 1.026$; i.e., $p_z$ increases linearly in $z$, while $r_z$ increases sublinearly in $z$. For these values of $p_z$ and $r_z$, sterility invades for double mating ($n = 2$) but not for single mating ($n = 1$). For *Figure 7B*, we have $p_0 = 0.8$, $p_{1/4} = 0.9$, $r_0 = 1$, $r_{1/4} = 1.013$, and $r_{1/2} = 1.026$; i.e., $p_z$ increases sublinearly in $z$, while $r_z$ increases linearly in $z$. For these values of $p_z$ and $r_z$, sterility invades for double mating ($n = 2$) but not for single mating ($n = 1$).

In *Figure 9*, we show the values of the parameters $r_{1/6}$ and $r_{1/4}$ for which non-reproductive workers can invade for double and triple mating. There are many possibilities. For example, it is possible that worker sterility evolves for triple mating but not for double or single mating. Sample forms of the functions $p_z$ and $r_z$ are shown in *Figure 10A,B*. For *Figure 10A*, we have $p_0 = 0.1$, $p_{1/6} = 0.25$, $p_{1/4} = 0.325$, $r_0 = 1$, $r_{1/6} = 1.095$, $r_{1/4} = 1.117$, and $r_{1/2} = 1.16$; i.e., $p_z$ increases linearly in $z$, while $r_z$ increases sublinearly in $z$. For these values of $p_z$ and $r_z$, sterility invades for triple mating ($n = 3$) but not for double mating ($n = 2$) or single mating ($n = 1$). For *Figure 10B*, we have $p_0 = 0.2$, $p_{1/6} = 0.5$, $p_{1/4} = 0.6$, $r_0 = 1$, $r_{1/6} = 1.04$, $r_{1/4} = 1.06$, and $r_{1/2} = 1.12$; i.e., $p_z$ increases sublinearly in $z$, while $r_z$ increases linearly in $z$. For these values of $p_z$ and $r_z$, sterility invades for triple mating ($n = 3$) but not for double mating ($n = 2$) or single mating ($n = 1$).

## Stability of a dominant worker sterility allele

We assume that a dominant worker sterility allele has spread to fixation. We consider the evolutionary stability of a population consisting entirely of sterile workers to invasion by reproductive workers. What happens if we start with an infinitesimal quantity of the mutant allele, $A$, by perturbing the $X_{aa,n} = 1$ pure equilibrium: $X_{aa,n} \longrightarrow 1 - \epsilon \delta^{(1)}_{aa,n}$, with $\epsilon \ll 1$? Does the dominant worker sterility allele return to fixation, or is it invaded by the worker reproduction allele?

Although the state space is ($3n + 2$)-dimensional ($3n + 3$ types of colonies subject to the density constraint), the analysis simplifies. Provided that the perturbation is small (i.e. that $\epsilon \ll 1$), only six colony types, $aa,n$, $aa,n-1$, $Aa,n$, $aa,n-2$, $Aa,n-1$, and $AA,n$, determine whether or not the dominant worker sterility allele is stable. Any other colony type is headed by a queen that possesses at least three mutant $A$ alleles (from her own genotype combined with the sperm she has collected), but such queens are so rare as to be negligible. The relevant equations among (*Equation 11*) for studying stability of a dominant sterility allele are

$$\begin{aligned}
\dot{X}_{aa,n} &= x_{aa}y_a^n - \phi X_{aa,n} \\
\dot{X}_{aa,n-1} &= n x_{aa}y_a^{n-1}y_A - \phi X_{aa,n-1} \\
\dot{X}_{Aa,n} &= x_{Aa}y_a^n - \phi X_{Aa,n} \\
\dot{X}_{aa,n-2} &= \frac{n(n-1)}{2} x_{aa}y_a^{n-2}y_A^2 - \phi X_{aa,n-2} \\[6pt]
\dot{X}_{Aa,n-1} &= n x_{Aa}y_a^{n-1}y_A - \phi X_{Aa,n-1} \\
\dot{X}_{AA,n} &= x_{AA}y_a^n - \phi X_{AA,n}
\end{aligned} \tag{35}$$

For analysis of the dominant sterility allele, terms of order $\epsilon$ do not provide sufficient information for determining whether the allele is stable, which adds a level of tedium to the calculation. Formally keeping track of powers of $\epsilon$ and $\epsilon^2$, and disregarding higher-order terms, we have:

$$
\begin{aligned}
X_{aa,n} &= 1 - \epsilon\delta_{aa,n}^{(1)} - \epsilon^2\delta_{aa,n}^{(2)} - \mathcal{O}(\epsilon^3) \\
X_{aa,n-1} &= +\epsilon\delta_{aa,n-1}^{(1)} + \epsilon^2\delta_{aa,n-1}^{(2)} + \mathcal{O}(\epsilon^3) \\
X_{Aa,n} &= +\epsilon\delta_{Aa,n}^{(1)} + \epsilon^2\delta_{Aa,n}^{(2)} + \mathcal{O}(\epsilon^3) \\
X_{aa,n-2} &= +\epsilon^2\delta_{aa,n-2}^{(2)} + \mathcal{O}(\epsilon^3) \\
X_{Aa,n-1} &= +\epsilon^2\delta_{Aa,n-1}^{(2)} + \mathcal{O}(\epsilon^3) \\
X_{AA,n} &= +\epsilon^2\delta_{AA,n}^{(2)} + \mathcal{O}(\epsilon^3)
\end{aligned}
\tag{36}
$$

To determine the density constraints, we substitute (*Equation 36*) into (*Equation 12*) and collect powers of $\epsilon$ and $\epsilon^2$. At order $\epsilon$, we get

$$
\delta_{aa,n}^{(1)} = \delta_{aa,n-1}^{(1)} + \delta_{Aa,n}^{(1)}
\tag{37}
$$

At order $\epsilon^2$, we get

$$
\delta_{aa,n}^{(2)} = \delta_{aa,n-1}^{(2)} + \delta_{Aa,n}^{(2)} + \delta_{aa,n-2}^{(2)} + \delta_{Aa,n-1}^{(2)} + \delta_{AA,n}^{(2)}
\tag{38}
$$

Next, we substitute (*Equation 36*) into (*Equation 14*), using the density constraints (*Equation 37*) and (*Equation 38*) and keeping terms up to order $\epsilon^2$:

$$
\begin{aligned}
x_{aa}r_1^{-1} &= 1 + \epsilon[\frac{-1}{n}\delta_{aa,n-1}^{(1)} - \frac{1}{2}\delta_{Aa,n}^{(1)}] + \epsilon^2[\frac{-1}{n}\delta_{aa,n-1}^{(2)} - \frac{1}{2}\delta_{Aa,n}^{(2)} \\
&\qquad -\frac{2}{n}\delta_{aa,n-2}^{(2)} + \frac{-2n + (n-1)r_{\frac{2n-1}{2n}}r_1^{-1}}{2n}\delta_{Aa,n-1}^{(2)} - \delta_{AA,n}^{(2)}] + \mathcal{O}(\epsilon^3) \\
x_{Aa}r_1^{-1} &= \epsilon[\frac{1}{n}\delta_{aa,n-1}^{(1)} + \frac{1}{2}\delta_{Aa,n}^{(1)}] + \epsilon^2[\frac{1}{n}\delta_{aa,n-1}^{(2)} + \frac{1}{2}\delta_{Aa,n}^{(2)} \\
&\qquad +\frac{2}{n}\delta_{aa,n-2}^{(2)} + \frac{r_{\frac{2n-1}{2n}}r_1^{-1}}{2}\delta_{Aa,n-1}^{(2)} + \delta_{AA,n}^{(2)}] + \mathcal{O}(\epsilon^3) \\
x_{AA}r_1^{-1} &= \epsilon^2[\frac{r_{\frac{2n-1}{2n}}r_1^{-1}}{2n}\delta_{Aa,n-1}^{(2)}] + \mathcal{O}(\epsilon^3) \\
y_a r_1^{-1} &= 1 + \epsilon[\frac{-1}{2}\delta_{Aa,n}^{(1)}] + \epsilon^2[\frac{-1}{2}\delta_{Aa,n}^{(2)} \\
&\qquad +\frac{-2 + p_{\frac{2n-1}{2n}}r_{\frac{2n-1}{2n}}r_1^{-1}}{2}\delta_{Aa,n-1}^{(2)} - \delta_{AA,n}^{(2)}] + \mathcal{O}(\epsilon^3) \\
y_A r_1^{-1} &= \epsilon[\frac{1}{2}\delta_{Aa,n}^{(1)}] + \epsilon^2[\frac{1}{2}\delta_{Aa,n}^{(2)} \\
&\qquad +\frac{2 - p_{\frac{2n-1}{2n}}}{2}r_{\frac{2n-1}{2n}}r_1^{-1}\delta_{Aa,n-1}^{(2)} + \delta_{AA,n}^{(2)}] + \mathcal{O}(\epsilon^3)
\end{aligned}
\tag{39}
$$

By plugging (*Equation 39*) and (*Equation 36*) into (*Equation 35*), using the density constraint (*Equation 37*), and collecting powers of $\epsilon$, we find

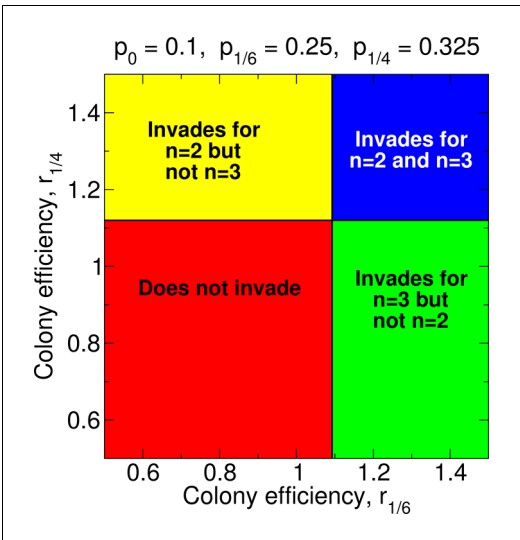

**Figure 9.** Regions of the parameter space and evolution of worker sterility for double and for triple mating. For double mating, $n = 2$, the invasion of a recessive worker sterility allele depends on the parameters $p_0$, $p_{1/4}$, and $r_{1/4}$; for triple mating, $n = 3$, it depends on the parameters $p_0$, $p_{1/6}$, and $r_{1/6}$. We set $r_0 = 1$ as baseline.

$$\dot{\delta}_{aa,n-1}^{(1)} = -r_1^{n+1}\delta_{aa,n-1}^{(1)} + \frac{n}{2}r_1^{n+1}\delta_{Aa,n}^{(1)}$$

$$\dot{\delta}_{Aa,n}^{(1)} = \frac{1}{n}r_1^{n+1}\delta_{aa,n-1}^{(1)} - \frac{1}{2}r_1^{n+1}\delta_{Aa,n}^{(1)}$$

The equations for $\dot{\delta}_{aa,n-1}^{(1)}$ and $\dot{\delta}_{Aa,n}^{(1)}$ can be written in matrix form as

$$\begin{pmatrix} \dot{\delta}_{aa,n-1}^{(1)} \\ \dot{\delta}_{Aa,n}^{(1)} \end{pmatrix} = r_1^{n+1} \begin{pmatrix} -1 & \dfrac{n}{2} \\ \dfrac{1}{n} & \dfrac{-1}{2} \end{pmatrix} \begin{pmatrix} \delta_{aa,n-1}^{(1)} \\ \delta_{Aa,n}^{(1)} \end{pmatrix}$$

The two eigenvectors ($v_0$ and $v_-$) and their corresponding eigenvalues ($\lambda_0$ and $\lambda_-$) are

$$v_0 = \begin{pmatrix} n \\ 2 \end{pmatrix} \quad \lambda_0 = 0$$

$$v_- = \begin{pmatrix} n \\ -1 \end{pmatrix} \quad \lambda_- = \frac{-3}{2}r_1^{n+1}$$

Notice that the dominant eigenvalue is equal to zero, so a computation to leading order in $\epsilon$ cannot provide information on the stability of the dominant sterility allele.

We can also see this more formally. An arbitrary initial perturbation to a resident $A$ population can be written as a linear superposition of the eigenvectors $v_0$ and $v_-$:

$$\begin{pmatrix} \delta_{aa,n-1}^{(1)} \\ \delta_{Aa,n}^{(1)} \end{pmatrix} = C_0 \begin{pmatrix} n \\ 2 \end{pmatrix} + C_- \begin{pmatrix} n \\ -1 \end{pmatrix} \exp\left(\frac{-3}{2}r_1^{n+1}t\right) \tag{40}$$

Here $C_0$ and $C_-$ are constants. We can substitute (*Equation 39*) and (*Equation 36*) into (*Equation 35*), using the density constraints (*Equation 37*) and (*Equation 38*), keeping terms of order $\epsilon$ and $\epsilon^2$, and dividing each term by one factor of $\epsilon$. We obtain

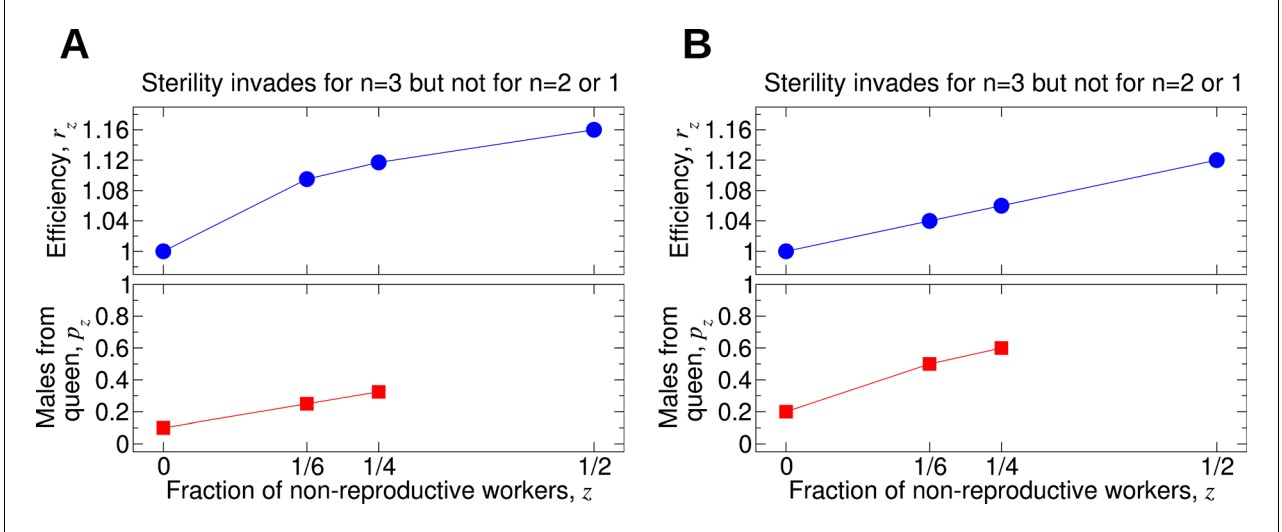

**Figure 10.** Comparing the effects of single mating ($n = 1$), double mating ($n = 2$), and triple mating ($n = 3$) on the evolution of worker sterility. Whether or not triple mating favors the evolution of worker sterility depends on the functions $p_z$ and $r_z$. We consider a recessive mutant allele, $a$, for worker sterility. (A, B) For these parameter choices, the mutant allele causing worker sterility can invade for triple mating but not for double or single mating.

$$
\begin{aligned}
[-\dot{\delta}^{(1)}_{aa,n} - \epsilon\dot{\delta}^{(2)}_{aa,n}]r_1^{-(n+1)} = {}& \frac{1-n}{2n}\left(-2\delta^{(1)}_{aa,n-1} + n\delta^{(1)}_{Aa,n}\right) \\
&+ \epsilon\left[\frac{1-n}{2n}\left(-2\delta^{(2)}_{aa,n-1} + n\delta^{(2)}_{Aa,n}\right)\right. \\
&+ \frac{n-2}{n}\delta^{(2)}_{aa,n-2} \\
&+ \frac{2n - \left(1 + n\left(1 + n\left(2 - \frac{p_{2n-1}}{2n}\right)\right)\right)\frac{r_{2n-1}r_1^{-1}}{2n}}{2n}\delta^{(2)}_{Aa,n-1} \\
&- n\delta^{(2)}_{AA,n} \\
&+ \frac{n(n+1)}{8}[\delta^{(1)}_{Aa,n}]^2 \\
&\left.+ \frac{1}{2}\delta^{(1)}_{aa,n-1}\delta^{(1)}_{Aa,n}\right]
\end{aligned}
\tag{41}
$$

We can again use the density constraints (*Equation 37*) and (*Equation 38*) to rewrite the left-hand side of (*Equation 41*). We can also substitute the general solution for the quantities $\delta^{(1)}_{aa,n-1}$ and $\delta^{(1)}_{Aa,n}$, *Equation 40*, into the right-hand side of (*Equation 41*):

$$[-\dot{\delta}^{(1)}_{aa,n-1} - \dot{\delta}^{(1)}_{Aa,n}]r_1^{-(n+1)}$$
$$+\epsilon[-\dot{\delta}^{(2)}_{aa,n-1} - \dot{\delta}^{(2)}_{Aa,n}$$
$$-\dot{\delta}^{(2)}_{aa,n-2} - \dot{\delta}^{(2)}_{Aa,n-1}$$

$$-\dot{\delta}^{(2)}_{AA,n}]r_1^{-(n+1)} \quad = \quad \frac{1-n}{2n}[-2\left(nC_0 + nC_-\exp\left(\frac{-3}{2}r_1^{n+1}t\right)\right)$$

$$+n\left(2C_0 - C_-\exp\left(\frac{-3}{2}r_1^{n+1}t\right)\right)]$$

$$+\epsilon[\frac{1-n}{2n}\left(-2\delta^{(2)}_{aa,n-1} + n\delta^{(2)}_{Aa,n}\right)$$

$$+\frac{n-2}{n}\delta^{(2)}_{aa,n-2}$$

$$+\frac{2n - \left(1 + n\left(1 + n\left(2 - \dfrac{p_{2n-1}}{2n}\right)\right)\right)\dfrac{r_{2n-1}r_1^{-1}}{2n}}{2n}\delta^{(2)}_{Aa,n-1}$$

$$-n\delta^{(2)}_{AA,n}$$

$$+\frac{n(n+1)}{8}[\delta^{(1)}_{Aa,n}]^2$$

$$+\frac{1}{2}\delta^{(1)}_{aa,n-1}\delta^{(1)}_{Aa,n}]$$

(42)

Note that each term in (*Equation 42*) involving the quantities $\delta^{(2)}_{aa,n-1}$, $\delta^{(2)}_{Aa,n}$, $\delta^{(2)}_{aa,n-2}$, $\delta^{(2)}_{Aa,n-1}$, and $\delta^{(2)}_{AA,n}$ is multiplied by $\epsilon$. In the limit $\epsilon \longrightarrow 0$, the quantities $\delta^{(2)}_{aa,n-1}$, $\delta^{(2)}_{Aa,n}$, $\delta^{(2)}_{aa,n-2}$, $\delta^{(2)}_{Aa,n-1}$, and $\delta^{(2)}_{AA,n}$ do not affect the dynamics of the quantities $\delta^{(1)}_{aa,n-1}$ and $\delta^{(1)}_{Aa,n}$. However, the quantities $\delta^{(1)}_{aa,n-1}$ and $\delta^{(1)}_{Aa,n}$ alone tell us nothing about whether or not the dominant sterility allele is stable against invasion by the mutant *A* allele. Therefore, we must consider the terms of order $\epsilon^2$ in our dynamical equations (*Equation 35*) to determine if the *a* allele is stable against invasion by the mutant *A* allele. In our calculations that follow, we use the eigenvector $v_0$ corresponding to the zero eigenvalue, i.e.

$$\begin{pmatrix} \delta^{(1)}_{aa,n-1} \\ \delta^{(1)}_{Aa,n} \end{pmatrix} = \frac{\delta^{(1)}_{aa,n}}{n+2}\begin{pmatrix} n \\ 2 \end{pmatrix}$$

(43)

Substituting (*Equation 39*), (*Equation 36*), and (*Equation 43*) into (*Equation 35*), using the density constraints (*Equation 37*) and (*Equation 38*), and keeping terms of order $\epsilon^2$, we have

$$
\begin{aligned}
-\dot{\delta}_{aa,n}^{(2)} r_1^{-(n+1)} =\ & \frac{1-n}{2n}\left(-2\delta_{aa,n-1}^{(2)} + n\delta_{Aa,n}^{(2)}\right) \\
& +\frac{n-2}{n}\delta_{aa,n-2}^{(2)} \\
& +\frac{2n-\left(1+n\left(1+n\left(2-p_{\frac{2n-1}{2n}}\right)\right)\right)r_{\frac{2n-1}{2n}}r_1^{-1}}{2n}\delta_{Aa,n-1}^{(2)} \\
& -n\delta_{AA,n}^{(2)} \\
& +\frac{n(n+3)}{2(n+2)^2}[\delta_{aa,n}^{(1)}]^2
\end{aligned}
\tag{44}
$$

We also have

$$
\begin{aligned}
\dot{\delta}_{aa,n-1}^{(2)} r_1^{-(n+1)} =\ & \frac{1}{2}\left(-2\delta_{aa,n-1}^{(2)} + n\delta_{Aa,n}^{(2)}\right) \\
& +\frac{n\left(2-p_{\frac{2n-1}{2n}}\right)}{2}r_{\frac{2n-1}{2n}}r_1^{-1}\delta_{Aa,n-1}^{(2)} \\
& +n\delta_{AA,n}^{(2)} - \frac{n(n+1)}{(n+2)^2}[\delta_{aa,n}^{(1)}]^2 \\[1ex]
\dot{\delta}_{Aa,n}^{(2)} r_1^{-(n+1)} =\ & \frac{-1}{2n}\left(-2\delta_{aa,n-1}^{(2)} + n\delta_{Aa,n}^{(2)}\right) \\
& +\frac{2}{n}\delta_{aa,n-2}^{(2)} \\
& +\frac{1}{2}r_{\frac{2n-1}{2n}}r_1^{-1}\delta_{Aa,n-1}^{(2)} \\
& +\delta_{AA,n}^{(2)} \\
& -\frac{2n}{(n+2)^2}[\delta_{aa,n}^{(1)}]^2 \\[1ex]
\dot{\delta}_{aa,n-2}^{(2)} r_1^{-(n+1)} =\ & -\delta_{aa,n-2}^{(2)} + \frac{n(n-1)}{2(n+2)^2}[\delta_{aa,n}^{(1)}]^2 \\[1ex]
\dot{\delta}_{Aa,n-1}^{(2)} r_1^{-(n+1)} =\ & -\delta_{Aa,n-1}^{(2)} + \frac{2n}{(n+2)^2}[\delta_{aa,n}^{(1)}]^2 \\[1ex]
\dot{\delta}_{AA,n}^{(2)} r_1^{-(n+1)} =\ & -\delta_{AA,n}^{(2)} + \frac{1}{2n}r_{\frac{2n-1}{2n}}r_1^{-1}\delta_{Aa,n-1}^{(2)}
\end{aligned}
$$

We can directly integrate the equation for $\dot{\delta}_{aa,n-2}^{(2)}$. We get

$$
\delta_{aa,n-2}^{(2)} = \frac{n(n-1)}{2(n+2)^2}[\delta_{aa,n}^{(1)}]^2[1-\exp(-r_1^{n+1}t)]
\tag{45}
$$

We can also directly integrate the equation for $\dot{\delta}_{Aa,n-1}^{(2)}$. We get

$$\delta_{Aa,n-1}^{(2)} = \frac{2n}{(n+2)^2}[\delta_{aa,n}^{(1)}]^2[1-\exp(-r_1^{n+1}t)] \tag{46}$$

We can use the solution for $\delta_{Aa,n-1}^{(2)}$ to solve for $\delta_{AA,n}^{(2)}$. We get

$$\delta_{AA,n}^{(2)} = \frac{r_{\frac{2n-1}{2n}}}{r_1(n+2)^2}[\delta_{aa,n}^{(1)}]^2[1-(1+r_1^{n+1}t)\exp(-r_1^{n+1}t)] \tag{47}$$

Manipulating the equations for $\dot\delta_{aa,n-1}^{(2)}$ and $\dot\delta_{Aa,n}^{(2)}$, we find that

$$r_1^{-(n+1)}\frac{d}{dt}(-2\delta_{aa,n-1}^{(2)}+n\delta_{Aa,n}^{(2)}) = \frac{-3}{2}\left(-2\delta_{aa,n-1}^{(2)}+n\delta_{Aa,n}^{(2)}\right)$$
$$+2\delta_{aa,n-2}^{(2)}$$
$$-\frac{n\left(3-2p_{\frac{2n-1}{2n}}\right)}{2}r_{\frac{2n-1}{2n}}r_1^{-1}\delta_{Aa,n-1}^{(2)}$$
$$-n\delta_{AA,n}^{(2)}$$
$$+\frac{2n}{(n+2)^2}[\delta_{aa,n}^{(1)}]^2$$

We can integrate this equation to solve for the quantity $-2\delta_{aa,n-1}^{(2)}+n\delta_{Aa,n}^{(2)}$. We obtain

$$-2\delta_{aa,n-1}^{(2)}+n\delta_{Aa,n}^{(2)} = [\frac{2n(n+1)}{3(n+2)^2}-\frac{2n\left(1+n\left(3-2p_{\frac{2n-1}{2n}}\right)\right)r_{\frac{2n-1}{2n}}}{3(n+2)^2r_1}][\delta_{aa,n}^{(1)}]^2$$
$$-[\frac{2n(n-1)}{(n+2)^2}$$
$$+\frac{2n\left(1-n\left(3-2p_{\frac{2n-1}{2n}}\right)-r_1^{n+1}t\right)r_{\frac{2n-1}{2n}}}{(n+2)^2r_1}][\delta_{aa,n}^{(1)}]^2\exp(-r_1^{n+1}t)$$
$$+[\frac{4n(n-2)}{3(n+2)^2}$$
$$+\frac{4n\left(2-n\left(3-2p_{\frac{2n-1}{2n}}\right)\right)r_{\frac{2n-1}{2n}}}{3(n+2)^2r_1}][\delta_{aa,n}^{(1)}]^2\exp\left(\frac{-3}{2}r_1^{n+1}t\right) \tag{48}$$

To determine if the resident $a$ population is unstable to invasion by the $A$ allele, we must consider the regime $1\ll t\ll\epsilon^{-1}$. Notice that on a short time scale, each of the time-dependent terms in *Equations 45–48* will approach zero. We must consider the sign of $\dot\delta_{aa,n}^{(2)}$ in the limit of large times $t\gg 1$ but before the terms in (*Equation 36*) become comparable in magnitude. Our condition for stability of the sterility allele is therefore

$$\lim_{\substack{\epsilon t \longrightarrow 0 \\ t \longrightarrow \infty}} \dot{\delta}^{(2)}_{aa,n} < 0 \tag{49}$$

Substituting (*Equations 44–48*) into (*Equation 49*), we find that the dominant allele for worker sterility is evolutionarily stable if

$$\frac{r_1}{r_{\frac{2n-1}{2n}}} > \frac{2+3n-np_{\frac{2n-1}{2n}}}{2(n+1)}$$

## Stability of a recessive worker sterility allele

We assume that a recessive worker sterility allele has spread to fixation. We consider the evolutionary stability of a population consisting entirely of sterile workers to invasion by reproductive workers. What happens if we start with an infinitesimal quantity of the mutant allele, $A$, by perturbing the $X_{aa,n} = 1$ pure equilibrium: $X_{aa,n} \longrightarrow 1 - \epsilon \delta^{(1)}_{aa,n}$, with $\epsilon \ll 1$? Does the recessive worker sterility allele return to fixation, or is it invaded by the worker reproduction allele?

Although the state space is $(3n + 2)$-dimensional ($3n + 3$ types of colonies subject to the density constraint), the analysis again simplifies. Provided that the perturbation is small (i.e. that $\epsilon \ll 1$), only three colony types, $aa,n$, $aa,n-1$, and $Aa,n$, determine whether or not the recessive worker sterility allele is evolutionarily stable. Any other colony type is headed by a queen that possesses at least two mutant $A$ alleles (from her own genotype combined with the sperm she has collected), but such queens are so rare as to be negligible. The relevant equations among (*Equation 11*) for studying stability of a recessive sterility allele are

$$\begin{aligned}
\dot{X}_{aa,n} &= x_{aa}y_a^n - \phi X_{aa,n} \\
\dot{X}_{aa,n-1} &= n x_{aa}y_a^{n-1}y_A - \phi X_{aa,n-1} \\
\dot{X}_{Aa,n} &= x_{Aa}y_a^n - \phi X_{Aa,n}
\end{aligned} \tag{50}$$

Formally keeping track of powers of $\epsilon$, and disregarding higher-order terms, we have:

$$\begin{aligned}
X_{aa,n} &= 1 - \epsilon \delta^{(1)}_{aa,n} - \mathcal{O}(\epsilon^2) \\
X_{aa,n-1} &= + \epsilon \delta^{(1)}_{aa,n-1} + \mathcal{O}(\epsilon^2) \\
X_{Aa,n} &= + \epsilon \delta^{(1)}_{Aa,n} + \mathcal{O}(\epsilon^2)
\end{aligned} \tag{51}$$

Next, we substitute (*Equation 51*) into (*Equation 15*), using the density constraint (*Equation 37*) and keeping terms only up to order $\epsilon$:

$$x_{aa} = r_1 + \varepsilon \left[ \frac{\frac{(n-1)r_{\frac{n-1}{n}} - nr_1}{n}}{n} \delta_{aa,n-1}^{(1)} + \frac{-2r_1 + r_{\frac{1}{2}}}{2} \delta_{Aa,n}^{(1)} \right] + \mathcal{O}(\varepsilon^2)$$

$$x_{Aa} = \varepsilon \left[ \frac{\frac{r_{\frac{n-1}{n}}}{n}}{n} \delta_{aa,n-1}^{(1)} + \frac{r_{\frac{1}{2}}}{2} \delta_{Aa,n}^{(1)} \right] + \mathcal{O}(\varepsilon^2)$$

$$y_a = r_1 + \varepsilon \left[ \frac{-2r_1 + \left(1 + p_{\frac{n-1}{n}}\right)r_{\frac{n-1}{n}}}{2} \delta_{aa,n-1}^{(1)} + \frac{-2r_1 + r_{\frac{1}{2}}}{2} \delta_{Aa,n}^{(1)} \right] + \mathcal{O}(\varepsilon^2) \tag{52}$$

$$y_A = \epsilon \left[ \frac{\left(1 - p_{\frac{n-1}{n}}\right)r_{\frac{n-1}{n}}}{2} \delta_{aa,n-1}^{(1)} + \frac{r_{\frac{1}{2}}}{2} \delta_{Aa,n}^{(1)} \right] + \mathcal{O}(\varepsilon^2)$$

By plugging (*Equation 52*) and (*Equation 51*) into (*Equation 50*), using the density constraint (*Equation 37*), and collecting powers of $\epsilon$, we find

$$\dot{\delta}_{aa,n-1}^{(1)} = \frac{-2r_1^{n+1} + n\left(1 - p_{\frac{n-1}{n}}\right)r_{\frac{n-1}{n}}r_1^n}{2} \delta_{aa,n-1}^{(1)} + \frac{nr_{\frac{1}{2}}r_1^n}{2} \delta_{Aa,n}^{(1)}$$

$$\dot{\delta}_{Aa,n}^{(1)} = \frac{r_{\frac{n-1}{n}}r_1^n}{n} \delta_{aa,n-1}^{(1)} + \frac{-2r_1^{n+1} + r_{\frac{1}{2}}r_1^n}{2} \delta_{Aa,n}^{(1)}$$

The equations for $\dot{\delta}_{aa,n-1}^{(1)}$ and $\dot{\delta}_{Aa,n}^{(1)}$ can be written in matrix form as

$$\begin{pmatrix} \dot{\delta}_{aa,n-1}^{(1)} \\ \dot{\delta}_{Aa,n}^{(1)} \end{pmatrix} = \begin{pmatrix} \dfrac{-2r_1^{n+1} + n\left(1 - p_{\frac{n-1}{n}}\right)r_{\frac{n-1}{n}}r_1^n}{2} & \dfrac{nr_1 r_1^n}{2} \\ \dfrac{r_{\frac{n-1}{n}}r_1^n}{n} & \dfrac{-2r_1^{n+1} + r_{\frac{1}{2}}r_1^n}{2} \end{pmatrix} \begin{pmatrix} \delta_{aa,n-1}^{(1)} \\ \delta_{Aa,n}^{(1)} \end{pmatrix}$$

Setting the dominant eigenvalue to be greater than zero and simplifying, we find that the recessive allele for worker sterility is evolutionarily stable if

$$\left[ \frac{r_1}{r_{\frac{n-1}{n}}} - \frac{n\left(1 - p_{\frac{n-1}{n}}\right)}{2} \right] \left[ 2\left(\frac{r_1}{r_{\frac{1}{2}}}\right) - 1 \right] > 1 \tag{53}$$

**Table 1.** Numerical experiments. We randomly select the two relevant colony efficiency values from a bivariate normal distribution. For Procedure 1, the two efficiency values are uncorrelated. For Procedure 2, they are correlated (with correlation 0.8). The results of the numerical experiment for *Figures 3A* and *8* are shown. For *Figure 3A*, which describes a recessive allele inducing non-reproductive workers, we randomly generate values for $r_{1/4}$ and $r_{1/2}$. For *Figure 8*, which describes a dominant allele inducing non-reproductive workers, we randomly generate values for $r_{1/2}$ and $r_1$. The table shows the likelihood of the four possible outcomes: non-reproductive workers (i) do not invade, (ii) invade for single mating but not for double mating, (iii) invade for double mating but not for single mating, and (iv) invade for both single and double mating. For this particular randomization experiment, double mating is more favorable than single mating for the invasion of non-reproductive workers. All *p* values are exactly as in the corresponding Figures.

|  | Does not invade | Invades for $n = 1$ but not $n = 2$ | Invades for $n = 2$ but not $n = 1$ | Invades for both $n = 1$ and $n = 2$ |
|---|---|---|---|---|
| *Figure 3A*, Proc. 1, recessive | 0.7769 | 0.0644 | 0.1465 | 0.0122 |
| *Figure 3A*, Proc. 2, recessive | 0.8237 | 0.0177 | 0.0997 | 0.0589 |
| *Figure 8*, Proc. 1, dominant | 0.7944 | 0.0129 | 0.0830 | 0.1097 |
| *Figure 8*, Proc. 2, dominant | 0.7927 | 0.0146 | 0.0260 | 0.1667 |

*Figure 3B* shows how the number of matings affects the evolutionary stability of non-reproductive workers for different values of the parameters $r_{1/2}$ and $r_1$. Sample forms of the functions $p_z$ and $r_z$ are shown in *Figure 7C,D*. For *Figure 7C*, we have $p_0 = 0.8$, $p_{1/2} = 0.92$, $r_0 = 1$, $r_{1/2} = 1.016$, and $r_1 = 1.045$; i.e., $p_z$ increases sublinearly in $z$, while $r_z$ increases superlinearly in $z$. For these values of $p_z$ and $r_z$, sterility is stable for double mating ($n = 2$) but not for single mating ($n = 1$). For *Figure 7D*, we have $p_0 = 0.8$, $p_{1/2} = 0.94$, $r_0 = 1$, $r_{1/2} = 1.0225$, and $r_1 = 1.045$; i.e., $p_z$ increases sublinearly in $z$, while $r_z$ increases linearly in $z$. For these values of $p_z$ and $r_z$, sterility is stable for double mating ($n = 2$) but not for single mating ($n = 1$).

## Numerical experiments

For additional insight, we perform random sampling of the parameter space to obtain some intuition whether evolution of non-reproductive workers is more or less likely for single or double mating. We will also evaluate the likelihood of selection favoring invasion or evolutionary stability of alleles (mutations) that induce non-reproductive workers. Thus, we do random sampling of the parameter regions shown in *Figures 3A*, *5A,* and *8*. In each case, the outcome depends on two efficiency values, which we call $r_{z1}$ and $r_{z2}$ with $z1 < z2$. For *Figure 3A*, those values are $r_{1/4}$ and $r_{1/2}$. For *Figure 5A* and for *Figure 8*, those values are $r_{1/2}$ and $r_1$.

The outcome of this numerical experiment depends on how we choose to randomize the colony efficiency values, $r_{z1}$ and $r_{z2}$. There are many ways to do this. Here, we consider two possibilities:

**Table 2.** Numerical experiments. With the equivalent Procedures, we explore the likelihood of the four scenarios regarding invasion and/or stability for single mating. Results of the numerical experiment for *Figure 5A*, describing a recessive allele, are shown. We randomly generate values for $r_{1/2}$ and $r_1$. The value $p_0 = 0.5$ is exactly as in *Figure 5A*.

|  | Does not invade and is unstable | Does not invade but is stable | Invades but is unstable | Invades and is stable |
|---|---|---|---|---|
| *Figure 5A*, Proc. 1, recessive | 0.3484 | 0.3014 | 0.3007 | 0.0495 |
| *Figure 5A*, Proc. 2, recessive | 0.5295 | 0.1203 | 0.2379 | 0.1123 |

- Procedure 1: We choose $r_{z1}$ and $r_{z2}$ from a bivariate normal distribution:

$$P(r_{z1}, r_{z2}) = \frac{1}{2\pi\sigma^2} \exp\left(\frac{-[(r_{z1} - \mu)^2 + (r_{z2} - \mu)^2]}{2\sigma^2}\right)$$

There is no correlation between $r_{z1}$ and $r_{z2}$. The average is $\mu = 1$. We choose $\sigma = 0.1$ for *Figure 3A*. We choose $\sigma = 0.2$ for *Figures 5A* and *8*.

- Procedure 2: We choose $r_{z1}$ and $r_{z2}$ from a bivariate normal distribution:

$$P(r_{z1}, r_{z2}) = \frac{1}{2\pi\sigma^2\sqrt{1-\rho^2}} \exp\left(\frac{-[(r_{z1} - \mu)^2 + (r_{z2} - \mu)^2 - 2\rho(r_{z1} - \mu)(r_{z2} - \mu)]}{2\sigma^2(1-\rho^2)}\right)$$

We set $\rho = 0.8$. Now, there is positive correlation between $r_{z1}$ and $r_{z2}$. We choose μ and σ as for Procedure 1.

*Table 1* shows the outcome of this numerical experiment for the parameter values used in *Figures 3A* and *8*. *Table 2* shows the outcome of this numerical experiment for the parameter values used in *Figure 5A*. For example, consider the first row of *Table 1*. We set $p_0 = 0.2$ and $p_{1/4} = 0.4$ with a recessive sterility allele, as this corresponds with *Figure 3A*. Procedure 1 is used for selecting values of $r_{1/4}$ and $r_{1/2}$. For a randomly chosen pair of efficiency values $r_{1/4}$ and $r_{1/2}$, the probabilities that the sterility allele does not invade, invades only for $n = 1$, invades only for $n = 2$, and invades for $n = 1$ and $n = 2$ are 0.7769, 0.0644, 0.1465, and 0.0122, respectively. For the second row of *Table 1*, Procedure 2 is used for selecting values of $r_{1/4}$ and $r_{1/2}$. For a randomly chosen pair of efficiency values $r_{1/4}$ and $r_{1/2}$, the probabilities that the sterility allele does not invade, invades only for $n = 1$, invades only for $n = 2$, and invades for $n = 1$ and $n = 2$ are 0.8237, 0.0177, 0.0997, and 0.0589, respectively. The third and fourth rows of *Table 1* and the rows of *Table 2* are understood in the same way.

For both Procedures, we find that the invasion of non-reproductive workers is more likely favored for double mating, $n = 2$, than for single mating, $n = 1$.

## Acknowledgements

We thank the referees and editor for helpful suggestions that have significantly benefited this manuscript.

## Additional information

### Funding

The authors declare that there was no funding for this work.

### Author contributions

JWO, BA, CV, MAN, Conception and design, Acquisition of data, Analysis and interpretation of data, Drafting or revising the article

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
