## [Decision Letter]

Thank you for submitting your work entitled "The evolution of non-reproductive workers in insect colonies with haplodiploid genetics" for peer review at *eLife*. Your submission has been favorably evaluated by Ian Baldwin (Senior Editor) and three reviewers, one of whom is a member of our Board of Reviewing Editors.

You will be pleased to see that the reviews are generally favorable, and we are willing to consider a revised submission that addresses the following concerns raised by the reviewers.

*Reviewer #1:* This is an interesting and potentially important paper that sheds new light on a classical problem. The authors show that in contrast to often made heuristic arguments, it is not true that monandry (single matings) of queens is necessary for the evolution of sterile workers, and hence eusociality. In fact, the authors show that depending on the fraction of males coming from the queen and on the benefit provided by sterile workers, multiple matings may sometimes be more conducive to the evolution of eusociality. The arguments are based on a population genetic model whose analysis appears to be sound. Given the importance of the results, which have the potential to generate a paradigm shift in the theory of eusociality, I recommend publication of this paper. Nevertheless, I have two major concerns that I think the authors should address prior to publication.

First, I think the Introduction is difficult to read for scientists who are not already familiar with the theory of eusociality. The term "haplodiploid" is used without explanation, and several logical steps need fleshing out. For example, if the eggs of a worker are unfertilized, why is relatedness between offspring only 1/2, and not 1? In general, various claims and statements about "relatedness" need to be better explained. Further, I don't understand the logic behind the "thus" in the seventh paragraph. I also don't understand last column (worker's sons) in Figure 1. This should be better explained. And I don't really understand why the quantity *p_z_*is not an emergent property of the model.

I also don't find the Discussion very convincing. For example, in the second paragraph the concept of worker policing is introduced seemingly out of the blue. On the one hand, the paragraph seems to discuss previous claims that monandry is necessary for eusociality, but then suddenly switches to talking about alternative causes of worker sterility (such as policing), which does not really seem pertinent and makes the logical flow somewhat unconvincing. All of the above needs serious attention if this paper is to be published in a journal such as *eLife*, which has a wide and general biology audience.

Second, the main claim that multiple matings may be more conducive to eusociality, and in fact may be necessary for the evolution of worker sterility, is based on a proof of principle: it is shown that there are cases (i.e., choices of parameter values for the model) in which this is true. However, I did not get a sense of how often these situations are expected to occur. I would like to see an analysis that reveals how "large" the regions in parameter space are for which multiple matings are more conducive to the evolution of eusociality than single matings, and how large the regions are for which multiple matings are necessary (i.e., single matings do not allow for eusociality to evolve, but multiple matings do). Such an analysis would give a quantitative sense of how large the paradigm shift generated by the results really is. Also, are there cases where more than two matings are necessary for the evolution of eusociality? If there are, this would be interesting to know as well.

*Reviewer #2:* The authors have constructed a very "flexible" model of the population dynamics of haplodiploid organisms, with alleles *A* and *a*, that live in large colonies. The model is studied via systems of (ordinary) differential equations. By analyzing limiting cases of the systems of ODEs they derive exact conditions for invasion and stability of a worker sterility mutation, *a*. Worker sterility is a defining feature of eusociality.

They argue convincingly that the functions *r_z_* (colony "efficiency") and *p_z_*(% of males from the queen):

1) Can (theoretically) take on a large number of qualitatively different forms;

2) Are not well understood even for modern species, and perhaps are unknowable in cases from the distant past;

3) Are crucial in determining whether or not behaviors like worker sterility can invade and/or stabilize.

Point 2 might seem to make a case against the proposed methodology, but it does not. The approach is honest, and ultimately much more satisfying than (paraphrasing the authors) "heuristics based on relatedness arguments".

I recommend this paper for publication in *eLife*, as I think the approach to eusociality taken is an important theoretical advance. I have a few comments that the authors may want to consider in a revision.

1) If I understand correctly, nobody knows what the functions *p_z_*and *r_z_* look like, aside from that they are (probably) increasing in z. What I take from Figure 3 and Figure 4, and the Discussion, is that subtle differences in *p_z_*and *r_z_* can lead to very different outcomes. But these figures give just a few examples out of an infinity of possibilities. To illustrate in a different way, I would suggest the following experiment – or something like it. Generate the functions *p_z_* and *r_z_* randomly, subject to the constraint that they are increasing. There are many ways to do this (actually, only a few values of the function are needed.) It is then a simple matter to decide (for *n*=1 and 2, and dominant/recessive) which of (+ or -) invasion and (+ or -) stability occur. The joint distribution for all possible outcomes can then be estimated by Monte Carlo. This would be very interesting (to me, anyway), and easy to do. Of course, someone could quibble with the way you generated *r_z_* and *p_z_*, but I think that is exactly the conversation you want to have.

2) Is there any rule of thumb for when *n*=2 is more conducive to worker sterility than *n*=1? Perhaps some insights would come from a statistical analysis of the simulation data from the previous comment.

3) I would be interested to see (numerical) solutions of the differential equations (5), perhaps corresponding to the examples in Figure 3 and Figure 4. I'm curious why the authors do not provide this, as the solutions would be very interesting, and also provide a check on the main results.

*Reviewer #3:* Olejarz et al. provide a mathematical model detailing the selective forces that determine the emergence and stability of non-reproductive workers. Understanding how the degree of genetic relatedness influences the evolution of eusociality is a key question in evolutionary biology. Recently, the role and significance of genetic relatedness as a factor facilitating the evolution eusociality has been hotly debated, and in this context, the authors’ conclusions would be quite controversial as their model attempts to show that monogamy (single queen that is singly mated), while increasing the genetic relatedness to the offspring, is not a necessary or facilitative condition, of eusociality. The mathematical model they present is quite elegant, but I have the following major concerns about the context and findings of the article:

1) Context: The major problem with the context of their findings is that the authors are equating the evolution eusociality with the evolution of non-reproductive workers. However, in ants, the evolution of eusociality occurs without the evolution of sterile, non-reproductive, workers. Workers in the basal lineages of ants are fully capable of mating and have the same number of ovarioles as the queen. The main difference in reproduction is in the overall activity of the ovarioles, and the queen regulates reproduction via policing. In fact, only 9 out of the ~300 genera of ants are completely sterile. So Darwin's famous quip that 'sterile' caste in ants possess a major challenge to his theory is misleading because sterility does not equate to origin of eusociality. This is not to say that the evolution of non-reproductive workers is not an important event in the history of eusociality in social insects, but it describes a transition to a much more advanced state of eusociality, long after the origin of eusociality occurs. Species with completely sterile workers are often ecological dominant (the fire ants) or evolutionarily successful (Pheidole, the big-headed ants). Therefore, the model the authors are proposing explains an important milestone in the social evolution, but is not reflective of the origin of the eusociality in most eusocial species. In this context, their model attempts to show that monogamy (single queen that is singly mated) is not necessarily a facilitating factor for the evolution of non-reproductive workers. They take this to mean that monogamy is then not a facilitating factor for the origin of eusociality. However, as I have explained above, they cannot equate evolution of eusociality to evolution of non-reproductive workers. The authors would have to re-write the manuscript and implications of their findings making the distinctions I have made above. Otherwise, the article will needlessly create controversy, and overlook the more interesting implications of their findings, especially in the light of molecular work (work from the Dearden and Abouheif labs) elucidating the molecular mechanisms of reproductive constraint in worker in bees and ants.

2) Model results: when comparing Figure 2 and Figure 3, it was initially unclear to me why the authors chose to model the sterility under a *p*_0_ = 0.5 and *p*_0_ = 0.9. According to Figure 2, *p*_0_ = 0.5 means that approximately 40-50% fraction of non-reproductive workers. Figure 3 shows that the conditions of colony efficiency under which a mutant allele can invade and is stable is relatively small and larger for *p*_0_ = 0.9. Therefore for *p*_0_'s less than 0.5 the conditions of colony efficiency under which a mutant allele can invade and be stable is very small. Therefore, it raises the question of how realistic, or likely, is it that the mutant allele will invade to lead to the evolution of non-reproductive workers. This may explain why it occurs so rarely in social insects (see above) and why eusociality is so rare to begin with. This is the much more interesting finding and the manuscript should be re-written to highlight these facts.

---

## [Author Response]

Reviewer #1:

*[…] First, I think the Introduction is difficult to read for scientists who are not already familiar with the theory of eusociality. The term "haplodiploid" is used without explanation, and several logical steps need fleshing out. For example, if the eggs of a worker are unfertilized, why is relatedness between offspring only 1/2, and not 1? In general, various claims and statements about "relatedness" need to be better explained. Further, I don't understand the logic behind the "thus" in the seventh paragraph. I also don't understand last column (worker's sons) in Figure 1. This should be better explained. And I don't really understand why the quantity* p_z_*is not an emergent property of the model.*

*I also don't find the Discussion very convincing. For example, in the second paragraph the concept of worker policing is introduced seemingly out of the blue. On the one hand, the paragraph seems to discuss previous claims that monandry is necessary for eusociality, but then suddenly switches to talking about alternative causes of worker sterility (such as policing), which does not really seem pertinent and makes the logical flow somewhat unconvincing. All of the above needs serious attention if this paper is to be published in a journal such as* eLife*, which has a wide and general biology audience.*

We fully agree that the Introduction and the descriptive text should be improved and targeted to a more general scientific audience. We have provided a better introduction to eusociality and have added text to highlight that the emergence of sterile workers is only one step in the evolution of advanced eusociality. We have added a description of the term “haplodiploid”. We provide a better explanation for our statements about relatedness.

The “thus” in the Introduction of the original submission refers to the fact that, in our model, if a worker does not lay an egg, then a queen-laid male egg takes its place. Therefore, worker-laid eggs (which are always male) are replaced by queen-laid male eggs, but not by queen-laid female eggs. This is why, in our model, the sex ratio is independent of the number of sterile workers. If, instead, queen-laid female eggs replace worker-laid male eggs, then the emergence of sterile workers would change the sex ratio of the colony. We have clarified this point.

We have added some descriptive text in the section “Model” regarding the “Workers’ Sons” columns in Figure 1. This should ease the reader’s interpretation of Figure 1. We also refer to the supplementary information for a more detailed description of Figure 1.

We have clarified some subtleties of the quantity *p_z_*, which is the fraction of male eggs that originate from the queen if a fraction, z, of workers are non-reproductive. One might initially think that *p_z_*would linearly increase in *z*. But this assumption is not necessarily true, because there can be nonlinear effects acting on the quantity *p_z_*. For example, the queen might remove worker-laid male eggs. If there are only a few reproducing workers (so that *z* is slightly less than 1), then the queen might effectively eliminate all worker-produced eggs. If there are many reproducing workers (so that *z* is slightly greater than 0), then the queen might only be able to remove a small fraction of all worker-laid eggs. In this case, *p_z_*would be expected to increase sublinearly with *z*.

Please note that *p_z_*is not an emerging property of the model. All we can say for sure is that *p_1_*=1. If all workers are non-reproductive, *z*=1, then all male eggs come from the queen. But if for example all of the workers are productive, *z*=0, then the value *p*_0_ does depend on biological details of that species, such as how many eggs are being laid by queen, how many by workers, how viable are the worker eggs, what happens to them, are they treated differently, etc.

We have rewritten the paragraph on policing. We have also added some text to the Discussion section to frame the phenomenon of worker sterility and other important aspects of eusociality in a clearer context. Our aim here is to make some statements about the broader discussion of evolution of eusociality.

*Second, the main claim that multiple matings may be more conducive to eusociality, and in fact may be necessary for the evolution of worker sterility, is based on a proof of principle: it is shown that there are cases (i.e., choices of parameter values for the model) in which this is true. However, I did not get a sense of how often these situations are expected to occur. I would like to see an analysis that reveals how "large" the regions in parameter space are for which multiple matings are more conducive to the evolution of eusociality than single matings, and how large the regions are for which multiple matings are necessary (i.e., single matings do not allow for eusociality to evolve, but multiple matings do). Such an analysis would give a quantitative sense of how large the paradigm shift generated by the results really is. Also, are there cases where more than two matings are necessary for the evolution of eusociality? If there are, this would be interesting to know as well.*

This is an excellent point. We have added Figure 3, Figure 8, and Figure 9, which show the regions of parameter space where non-reproductive workers invade for n+1 matings but not for n matings. An interesting principle emerges. When comparing *n*=1 and *n*=2 matings, increasing *r*_1/4_ favors evolution of non-reproductive workers for *n*=2 matings, while increasing *r*_1/2_ and leaving everything else constant favors evolution for *n*=1 matings. Additionally, we have added Figure 10, which illustrates sample forms of *p_z_*and *r_z_* for which non-reproductive workers invade for triple mating but not for double or single mating.

Reviewer #2:

*[…] I recommend this paper for publication in* eLife*, as I think the approach to eusociality taken is an important theoretical advance. I have a few comments that the authors may want to consider in a revision.*

*1) If I understand correctly, nobody knows what the functions* p_z_
*and* r_z_
*look like, aside from that they are (probably) increasing in z. What I take from Figure 3 and Figure 4, and the Discussion, is that subtle differences in* p_z_*and* r_z_
*can lead to very different outcomes. But these figures give just a few examples out of an infinity of possibilities. To illustrate in a different way, I would suggest the following experiment – or something like it. Generate the functions* p_z_*and* r_z_
*randomly, subject to the constraint that they are increasing. There are many ways to do this (actually, only a few values of the function are needed). It is then a simple matter to decide (for* n *= 1 and 2, and dominant/recessive) which of (+ or -) invasion and (+ or -) stability occur. The joint distribution for all possible outcomes can then be estimated by Monte Carlo. This would be very interesting (to me, anyway), and easy to do. Of course, someone could quibble with the way you generated* r_z_
*and* p_z_*, but I think that is exactly the conversation you want to have.*

This is an excellent suggestion. We have added Figure 3, Figure 8, and Figure 9, which show typical regions of the parameter space and the associated behavior of the sterility allele. We agree that the case of a monotonically increasing efficiency function *r_z_* is interesting, and we do not rule out the possibility of a function *r_z_* that reaches a maximum for an intermediate value of 0<z<1. For example, it may be that the addition of sterile workers results in diminishing returns in colony productivity. We have added text on these subtleties at the end of the Model section and also in the Results section.

We have also done some numerical experiments, as you suggested. In the subsection “Numerical Experiments” in the Methods section, we generate efficiency parameters randomly according to a bivariate normal distribution. We investigate both the cases where efficiency parameters are uncorrelated and also where efficiency parameters are positively correlated. These experiments were run for each of the scenarios depicted in Figure 3, Figure 5, and Figure 8, and the results are shown in Table 1.

*2) Is there any rule of thumb for when* n*=2 is more conducive to worker sterility than* n*=1? Perhaps some insights would come from a statistical analysis of the simulation data from the previous comment.*

Thank you for this excellent question, which led us to the following realization. The condition for non-reproductive workers to invade for single mating depends only on the parameters *p*_0_ and *r*_1/2_. The condition for them to invade with double mating depends only on the parameters *p*_0_, *p*_1/4_ and *r*_1/4_.

Here is one rule of thumb: Holding all other parameters constant, an increase in *r*_1/2_ favors the evolution of non-reproductive workers with single mating. Holding all other parameters constant, an increase in *r*_1/4_ favors the evolution of non-reproductive workers with double mating. We have added text on these points in the Results section.

Figure 3 and Figure 9 in the new version illustrate the parameter space for a recessive allele. These figures provide additional intuition for the various regions of parameter space.

For many values of *p*_0_ and *p*_1/4,_we find that *r*_1/4_ can be less than *r*_1/2_ (for example, *r_z_* might increase linearly or sublinearly with *z*), and sterility can be favored with double mating but not with single mating. We have shown some examples of this phenomenon in Figure 7 of the new version. The new Figure 10 shows examples for which sterility is favored with triple mating but not with single or double mating. We have also added some descriptive text on this point in the paper.

Figure 8 in the new version illustrates the parameter space for a dominant sterility allele. In the Methods section, where Figure 8 is referenced, we add a description of the parameter space for a dominant sterility allele.

*3) I would be interested to see (numerical) solutions of the differential equations (5), perhaps corresponding to the examples in Figure 3 and Figure 4. I'm curious why the authors do not provide this, as the solutions would be very interesting, and also provide a check on the main results.*

This is a great point. We have added numerical simulations of the evolutionary dynamics in the new Figure 4 and Figure 6.

Reviewer #3:

*Olejarz et al. provide a mathematical model detailing the selective forces that determine the emergence and stability of non-reproductive workers. Understanding how the degree of genetic relatedness influences the evolution of eusociality is a key question in evolutionary biology. Recently, the role and significance of genetic relatedness as a factor facilitating the evolution eusociality has been hotly debated, and in this context, the authors’ conclusions would be quite controversial as their model attempts to show that monogamy (single queen that is singly mated), while increasing the genetic relatedness to the offspring, is not a necessary or facilitative condition, of eusociality. The mathematical model they present is quite elegant, but I have the following major concerns about the context and findings of the article: 1) Context: The major problem with the context of their findings is that the authors are equating the evolution eusociality with the evolution of non-reproductive workers. However, in ants, the evolution of eusociality occurs without the evolution of sterile, non-reproductive, workers. Workers in the basal lineages of ants are fully capable of mating and have the same number of ovarioles as the queen. The main difference in reproduction is in the overall activity of the ovarioles, and the queen regulates reproduction via policing. In fact, only 9 out of the ~300 genera of ants are completely sterile. So Darwin's famous quip that 'sterile' caste in ants possess a major challenge to his theory is misleading because sterility does not equate to origin of eusociality. This is not to say that the evolution of non-reproductive workers is not an important event in the history of eusociality in social insects, but it describes a transition to a much more advanced state of eusociality, long after the origin of eusociality occurs. Species with completely sterile workers are often ecological dominant (the fire ants) or evolutionarily successful (Pheidole, the big-headed ants). Therefore, the model the authors are proposing explains an important milestone in the social evolution, but is not reflective of the origin of the eusociality in most eusocial species. In this context, their model attempts to show that monogamy (single queen that is singly mated) is not necessarily a facilitating factor for the evolution of non-reproductive workers. They take this to mean that monogamy is then not a facilitating factor for the origin of eusociality. However, as I have explained above, they cannot equate evolution of eusociality to evolution of non-reproductive workers. The authors would have to re-write the manuscript and implications of their findings making the distinctions I have made above. Otherwise, the article will needlessly create controversy, and overlook the more interesting implications of their findings, especially in the light of molecular work (work from the Dearden and Abouheif labs) elucidating the molecular mechanisms of reproductive constraint in worker in bees and ants.*

We fully agree. We did not mean to confuse the evolution of non-reproductive workers with the origin of eusociality. In the revised version of the manuscript, we fully clarify this difference, and we list the important points that you are making. We now describe evolution of non-reproductive (or sterile) workers as an important step in the evolution of advanced eusociality.

We have rewritten the manuscript accordingly. In the Discussion section, we have added text that explicitly acknowledges that worker sterility is only one aspect of advanced eusociality. We have added citations to interesting studies on other key aspects of eusociality, including nest formation, queen policing, and worker policing. We have also added citations to several important studies from both the Dearden and Abouheif labs that connect well with our work.

*2) Model results: when comparing Figure 2 and Figure 3, it was initially unclear to me why the authors chose to model the sterility under a* p*_0_ = 0.5 and* p*_0_ = 0.9. According to Figure 2*p*_0_ = 0.5 means that approximately 40-50% fraction of non-reproductive workers. Figure 3 shows that the conditions of colony efficiency under which a mutant allele can invade and is stable is relatively small and larger for* p*_0_ = 0.9. Therefore for* p*_0_'s less than 0.5 the conditions of colony efficiency under which a mutant allele can invade and be stable is very small. Therefore, it raises the question of how realistic, or likely, is it that the mutant allele will invade to lead to the evolution of non-reproductive workers. This may explain why it occurs so rarely in social insects (see above) and why eusociality is so rare to begin with. This is the much more interesting finding and the manuscript should be re-written to highlight these facts.*

Thank you for highlighting this important point.

We have made appropriate changes to the main text. After Equation (1), we now describe that small values of *p*_0_ (the fraction of male eggs from the queen when no workers are sterile) require efficiency increases of 10-20%. After Equation (2), we now also describe that large efficiency gains (of 10-20%) are needed.

Several paragraphs below, we now mention again that efficiency gains of 10-20% are large and may be difficult to achieve. Based on your suggestion, we conjecture that these large thresholds may aid in explaining the rarity of non-reproductive castes in the social insects.

We have added Figure 3, Figure 8, and Figure 9. These new figures show the regions of parameter space for which non-reproductive workers are favored by natural selection. We use small values of *p*_0_ in these new plots (*p*_0_=0.1 or 0.2) to illustrate that large efficiency thresholds are needed for sterility to develop.

In the second-to-last paragraph of the Discussion section, we mention again that large efficiency gains are needed for sterility to invade if *p*_0_ is small. We point out that this is a central result. We mention that larger values of *p*_0_ probably correspond with a higher degree of social organization already being established before sterility invades, such as queen policing or worker policing.